# Actor-only and Safe-Actor-only REINFORCE Algorithms with Deterministic Update Times

## Abstract

Regular Monte-Carlo policy gradient reinforcement learning (RL) algorithms require aggregation of data over regeneration epochs, constituting an episode (until a termination state is reached). In real-world applications involving large state and action spaces, the hitting times for goal states can be very sparse or infrequent, resulting in large episodes of unpredictable length. As an alternative, we present an RL algorithm called Actor-only algorithm (AOA) that performs data aggregation over a certain (deterministic) number of epochs. This helps remove unpredictability in the data aggregation step and thereby the update instants. Note also that satisfying safety constraints in RL is extremely crucial in safety-critical applications. We also extend the aforementioned AOA to the setting of safe RL that we call Safe-Actor-only algorithm (SAOA). In this work, we provide the asymptotic and finite-time convergence guarantees of our proposed algorithms to obtain the optimal policy. The finite-time analysis of our proposed algorithms demonstrates that finding a first-order stationary point, i.e., $\left\|\nabla \bar{J}(\theta)\right\|_2^2 \leq \epsilon$ and $\left\|\nabla \bar{\mathcal{L}}(\theta, \eta)\right\|_2^2 \leq \epsilon$ of performance function $\bar{J}(\theta)$ and $\bar{\mathcal{L}}(\theta, \eta)$, respectively, both with $\mathcal{O}(\epsilon^{-2})$ sample complexity. Further, our empirical results on benchmark RL environments demonstrate the advantages of proposed algorithms over considered algorithms in the literature.

## 1 Introduction

Reinforcement learning (RL) is a sequential decision-making paradigm that aims at finding the optimal sequence of actions in order to minimize or maximize a certain long-term objective when the system model of the underlying Markov decision process (MDP) (Puterman, 2014) is not known (Sutton & Barto, 2018). RL algorithms learn from data received from either a simulation device or a real source. RL has found applications in diverse domains including power systems, natural language processing, asset management, and robotics (Hakobyan et al., 2019). RL algorithms can broadly be classified as value-based, such as Q-learning, and policy-based methods, such as policy-gradient (PG) (Sutton & Barto, 2018). PG methods are often appropriate for high-dimensional state-action settings (Sutton et al., 2000) that occur in real-world applications. Among PG methods, actor-critic (AC) (Konda & Tsitsiklis, 2000) and soft actor-critic (SAC) (Haarnoja et al., 2018) are popular. PG algorithms update policy either at every time step or at the end of the trajectory (using the Monte Carlo PG (MCPG) (Noorani & Baras, 2021)), that is, when the goal state is reached. In this paper, we first present an Actor-only algorithm (AOA) that works with a single update recursion (instead of two recursions commonly used in the AC method) and works with linear function approximation. Further, our methodology updates the actor parameter after increasing deterministic instants using a Simultaneous Perturbation Stochastic Approximation (SPSA) based approach. Our proposed method is thus analogous to a trajectory-based method where the trajectories are of deterministic though increasing lengths.

We further extend our framework to incorporate inequality constraints obtained from single-stage costs in addition to rewards, and present a Safe-Actor-only algorithm (SAOA) that derives the optimal policy within a safe region. There is a lot of research activity on Safe RL in recent times and (García & Fernández, 2015) provides an overview of the same. Note again that regular RL aims to optimize the agent's performance via

its long-term reward, and in the process, learn an optimal policy interacting with the dynamic environment. This often requires significant exploration, which can often be unsafe in real-world applications.

In recent times, RL algorithms have been applied in several safety-critical applications, such as robotics, autonomous driving, cybersecurity, and financial management, where agents' safety is crucial (Kiran et al., 2022; Machado et al., 2017). Thus, the agent's goal here is not only to maximize long-term reward or achieve optimal policy but also to ensure that the agent never enters unsafe states, i.e., the agent must look for optimal solutions under safety constraints.

For our Safe-Actor-only algorithm, we introduce a safe PG method, where the underlying setting is a constrained Markov decision process (CMDP) (Altman, 1999),(Bhatnagar & Lakshmanan, 2012). Here, the constraint region can be designed to ensure the agent's safety. Our Safe-Actor-only method adds constraint functions to the original objective function, partitioning the state space into safe and unsafe regions. Our specific contributions are listed below.

- Our algorithms resolve the unpredictability of parameter update that lies in the Monte-Carlo PG methods. Our proposed algorithms perform parameter updates after increasing deterministic instants, unlike the regular Monte-Carlo PG methods, where the update happens after the end of each episode (episode lengths are random, not deterministic).

- We consider two different MDP settings - with and without constraints, and propose RL algorithms for both settings. Specifically, we propose (i) AOA for the regular MDP and (ii) SAOA for the constrained MDP, in the long-run average reward with function approximation settings.

- Both of our algorithms, AOA (involves two timescales) and SAOA (involves three timescales), are model-free RL algorithms and, in fact, versions of PG and constrained PG algorithms, respectively.

- We provide the convergence guarantees of our proposed algorithms by performing both asymptotic and non-asymptotic or finite-time analysis. We show that our algorithms in a non-i.i.d (Markovian) setting are guaranteed to converge to an $\epsilon$-neighborhood of the first-order stationary point, i.e., $\|\nabla \bar{J}(\theta)\|_2^2 \leq \epsilon$ and $\|\nabla \bar{\mathcal{L}}(\theta, \eta)\|_2^2 \leq \epsilon$ of the performance function $\bar{J}(\theta)$ and $\bar{\mathcal{L}}(\theta, \eta)$, respectively, with a sample complexity of $\mathcal{O}(\epsilon^{-2})$ for both algorithms AOA and SAOA, respectively. To the best of our knowledge, there are no other two-timescale and three-timescale algorithms that provide this sample complexity while remaining almost surely stable and convergent.

- We also provide empirical results, including regular and safe navigation of the RL agent in different standard environments, that demonstrate the effectiveness of the theoretical results.

## 2  Related Work

**PG for Regular MDPs:** PG algorithms are data-driven approaches and involve either trajectory-based methods or else are incremental update approaches. The latter fall under the broad category of actor-critic methods, while the former approaches are typically actor-only methods. We now go over some of the works that employ PG approaches.

PG methods with compatible function approximators are discussed in (Sutton et al., 2000). AC algorithms with PG actors have been studied and analyzed for their asymptotic convergence in (Konda & Tsitsiklis, 2000; Bhatnagar et al., 2009). In (Bhatnagar & Kumar, 2004; Abdulla & Bhatnagar, 2007), actor-critic algorithms for the look-up table setting and for the discounted and average (avg.) cost MDPs, respectively, are presented. These involve temporal difference (TD) learning critic and PG actor, where the actor update is based on simultaneous perturbation stochastic approximation (SPSA) (Spall, 1992) gradient estimates. In (Kumar & *et al.*, 2024; Qiu et al., 2019; Wu et al., 2020), AC algorithms are discussed where finite-time analysis (FTA) is done, but asymptotic analysis is not shown. In contrast, in (Mandal et al., 2024; Bhatnagar et al., 2009), asymptotic analysis is shown, but FTA is not available. In (Li et al., 2024), stochastic first-order methods for avg. reward MDPs are presented, and FTA is provided for the generative model and the Markovian model settings. In this work, parameter update depends on the aggregation of data until the termination of the trajectory, and can lead to infrequent or unpredictable parameter updates as trajectory

length is random, not deterministic. Furthermore, in (Li et al., 2024), the constrained MDP setting is not considered, and asymptotic convergence analysis is not demonstrated.

**PG for Constrained MDPs:** Incorporating safety constraints in RL is crucial for safety critical applications. The CMDP (Altman, 1999) is a widely studied framework for RL with constraints. It is assumed here that in addition to single-stage rewards, each state transition also fetches a set of single-stage costs that describe the (long-term) constraint functions. Constrained policy optimization procedures are based on this formulation, see for instance, (Achiam et al., 2017), (Tessler et al., 2019) and (D. Ding et al., 2020).

In (Borkar, 2004), a multi-timescale constrained AC algorithm in the look-up table case, for the long-run avg. reward criterion is presented. The Lagrange multiplier approach is adopted resulting in a three timescale algorithm. In addition to the actor and critic updates on different timescales, the Lagrange parameter is updated on the slowest timescale. Extending the idea in (Borkar, 1997), (Bhatnagar, 2010a) presents a constrained AC algorithm with function approximation for the discounted reward setting. The actor update here incorporates SPSA based gradient search.

In (Bhatnagar & Lakshmanan, 2012), an online constrained AC (CAC) algorithm with function approximation for the avg. reward setting is presented and asymptotic analysis is done. The actor update here incorporates the PG estimator ((Sutton et al., 2000)) for the Lagrangian obtained from relaxing the constraints in the objective. Model-based RL for constrained MDP is discussed in (Singh et al., 2023; A. H. Zonuzy & Shakkottai, 2021) for finite and infinite time horizon settings, respectively. Safe RL-based approaches are also discussed in (Ge et al., 2019; Wachi & Sui, 2020). In (Panda & Bhatnagar, 2024) CAC is discussed and FTA is shown. Table 1 summarizes the comparison of our proposed works, AOA and SAOA, with corresponding related works, in terms of sample complexity and asymptotic analysis.

## 3 Background

**Regular RL:** Markov Decision Process (MDP) is the backbone of regular RL. By an MDP, we mean a four-tuple $(\mathcal{S}, \mathcal{A}, r, P)$, where $\mathcal{S}, \mathcal{A}, r, P$ denote the state space, the action space, the reward function, and the probability transition matrix, respectively. We assume here finite state and action spaces. By a policy $\pi$, we mean a mapping $\pi : \mathcal{S} \to \Delta(\mathcal{A})$ from the state space $\mathcal{S}$ to the set of distributions over feasible actions in state $s \in \mathcal{S}$. In this work, we consider the average (avg.) reward setting and the aim is to find a policy $\pi^*$ that maximizes the long-run avg. reward, $J_\pi$, as follows:

$$\pi^* = \arg\max_{\pi \in \Pi} J_\pi, \text{ where } J_\pi = \lim_{n \to \infty} \frac{1}{n} \mathbb{E}[\sum_{k=0}^{n-1} r_k \mid \pi]. \tag{1}$$

We consider a class of policies $\pi^\theta$, parameterized by $\theta \in \mathbb{R}^d$, $d \geq 1$. Our objective then is to determine the optimal value of $\theta$ to maximize the long-run avg. reward $J_{\pi^\theta}$.

When the closed form of $\nabla_\theta J_{\pi^\theta}$ is known, one can find the optimal $\theta$ iteratively by following the gradient ascent scheme:

$$\theta_{n+1} = \theta_n + \alpha_n \nabla_\theta J_{\pi^\theta}, \ n \geq 0, \tag{2}$$

starting from an arbitrarily chosen $\theta_0 \in \mathbb{R}^d$ and $\alpha_n, n \geq 0$, is the step-size sequence. Since $\nabla_\theta J_{\pi^\theta}$ is not known, we adopt a novel stochastic gradient search-based procedure.

**Constrained RL:** Constrained RL algorithms such as constrained actor-critic (CAC) algorithms (Bhatnagar, 2010b; Bhatnagar & Lakshmanan, 2012) have found significant applications in the area of safe RL. Let $r_n$ denote the single-stage reward obtained at the $n$th instant as before. However, we shall assume that each state transition also fetches $N$ other single-stage costs $g_1(n), \ldots, g_N(n)$ at instant $n \geq 0$. Given the current state-action pair $(s_n, a_n)$, $r_n, g_q(n), q = 1, \ldots, N$ are assumed conditionally independent of the previous states and actions $(s_m, a_m, m < n)$. Further, $r(s, a)$ and $g_q(s, a)$ are defined as $r(s, a) := \mathbb{E}[r_n \mid s_n = s, a_n = a]$ and $g_q(s, a) := \mathbb{E}[g_q(n) \mid s_n = s, a_n = a], q = 1, \ldots, N$, respectively. Let, $d_\pi = (d_\pi(s), s \in \mathcal{S})$ be the stationary probability distribution of the ergodic Markov process $\{\mathcal{X}_n, n \geq 0\}$ (see Assumption 1). The objective is to

maximize the long-run avg. reward, given by

$$J_\pi = \lim_{n\to\infty} \frac{1}{n} \mathbb{E}\left[\sum_{k=0}^{n-1} r_k \mid \pi\right] = \sum_{s\in S} d_\pi(s) \sum_{a\in\mathcal{A}(s)} \pi(s,a) r(s,a). \tag{3}$$

This is, however done subject to the constraints

$$\mathcal{G}_q(\pi) := \lim_{n\to\infty} \frac{1}{n} \mathbb{E}\left[\sum_{q=0}^{n-1} g_q(k) \mid \pi\right] = \sum_{s\in S} d_\pi(s) \sum_{a\in\mathcal{A}(s)} \pi(s,a) g_q(s,a) \leq \nu_q, \tag{4}$$

$q = 1, \ldots, N$, where $\nu_1, \ldots, \nu_N$ are prescribed positive thresholds. The above problem that is defined in (3) and (4) is redefined using Lagrange relaxation as follows:

$$\mathcal{L}(\pi, \eta) := J_\pi - \sum_{q=1}^{N} \eta_q \left(\mathcal{G}_q(\pi) - \nu_q\right) = \sum_{s\in S} d_\pi(s) \sum_{a\in\mathcal{A}(s)} \pi(s,a) \left(r(s,a) - \sum_{q=1}^{N} \eta_q \left(g_q(s,a) - \nu_q\right)\right), \tag{5}$$

where, $\eta = (\eta_1, \ldots, \eta_N)^\top$ is a vector of Lagrange multipliers $\eta_q \in \mathbb{R}^+ \cup \{0\}, q = 1, \ldots, N$, with $\mathcal{L}(\pi, \eta)$ being the Lagrangian. We now consider at instant $n$, the single-stage reward for the relaxed problem as $r_n - \sum_{q=1}^{N} \eta_q \left(g_q(n) - \nu_q\right)$. In our work, we use Lagrangian and adopt a novel stochastic gradient search-based approach to get the optimal policy.

## 4 Proposed Algorithm

This section describes our proposed methodology, Actor-only (i.e., Algorithm 1) and Safe-Actor-only (i.e., Algorithm 2) REINFORCE algorithms with deterministic update times that resolve the unpredictability of parameter update lies in the Monte-Carlo PG methods.

### 4.1 Actor-only Algorithm (AOA) for Regular RL

---
**Algorithm 1** Actor-only Algorithm for Regular RL

---
**Input:** Scalar $\delta > 0$ and $\Delta$, a zero-mean, $\pm 1$-valued, Bernoulli distributed sample.
**Output:** Optimal policy
1: Initialisation : $\theta(0) = \theta_0$, $n_0 = 0$, $m = 0$, $0 < \sigma'' < \sigma''' \leq 1$, $J = 0$.
2: **for** $n = 0$ to $\infty$ **do**
3:     $\alpha_n = \frac{1}{\{1+n\}^{\sigma'''}}$, $\beta_n = \frac{1}{\{1+n\}^{\sigma''}}$ satisfy $\alpha_n < \beta_n$
4:     $n_{m+1} = \min\{j \geq n_m \mid \sum_{i=n_m+1}^{j} \alpha_i \geq \beta_{n_m}\}$
5:     Get next state $s'$, reward $r(n)$ using current state $s$ and action $a \sim \pi^\theta$.
6:     Get next state $s^+$, reward $r^+(n)$ using current state $s$ and action $a \sim \pi^{\theta+\delta\Delta}$.
7:     **if** $n == n_{m+1}$ **then**
8:         **for** $i = 1, \ldots, d$ **do**
9:             $\theta_i(m+1) = \Lambda_i[\theta_i(m) + (\sum_{j=n_m+1}^{n_{m+1}} \alpha_j \frac{(r^+(j)-r(j))}{\delta\Delta_i(m)})]$
10:        **end for**
11:        $m \leftarrow m + 1$
12:    **end if**
13:    $J(n+1) \leftarrow J(n) + \beta_n(r(n) - J(n))$
14:    Update the current state $s$ as the next state $s'$, i.e., $s \leftarrow s'$
15: **end for**
16: **return** Optimal policy parameter $\theta^*$.

---

Our proposed AOA (i.e., Algorithm 1) obtains the optimal policy in the long-run average reward setting. This algorithm computes the optimal policy parameter $\theta$, where the objective is to maximize the long-run

average reward. In this algorithm, we estimate the gradient of the objective function using the SPSA-based gradient estimates (Spall, 1992) as these are easy to compute and are found to be efficient. Here, we make the following standard assumptions as in the literature (Bhatnagar & Lakshmanan, 2012; Wu et al., 2020).

**Assumption 1.** *The Markov chain $\{\mathcal{X}_n, n \geq 0\}$ under any policy $\pi$ is ergodic.*

**Assumption 2.** *For any $a \in \mathcal{A}(s), s \in \mathcal{S}, \pi(s, a)$ is twice continuously differentiable in the policy parameter.*

By Assumption 1, the Markov chain is irreducible, aperiodic, and positive recurrent (Li et al., 2024). Under this assumption, the Markov chain has a unique stationary distribution $d_\pi$. We employ linear soft-max policy parameterizations (see eq. (6)) as we consider linear function approximation in this work.

$$\pi^\theta(s, a) = \frac{e^{\theta^\top \phi(s,a)}}{\sum_{a' \in A} e^{\theta^\top \phi(s,a')}}, \forall s \in \mathcal{S}, a \in \mathcal{A}, \tag{6}$$

where $\phi(s, a)$ is the state-action feature vector. Let, the set of parameterized policies be denoted $\tilde{\Pi}$, i.e., $\tilde{\Pi} = \{\pi^\theta | \theta \in \mathbb{R}^d\}$. Now, the optimization of policy parameter $\theta$ is only for policies in $\tilde{\Pi}$.

The proposed algorithm has a single update recursion, update of parameter $\theta$, that involves step-size $\alpha_n, n \geq 0$. The update epochs, $n_m \geq 0$, or alternatively $m \geq 0$ are obtained from two sets of step-sizes $\alpha_n, n \geq 0$, and $\beta_n, n \geq 0$, respectively. These step-size sequences satisfy (28)–(30) shown in the Appendix A. We take $\delta > 0$ as a small constant and assume that policy parameter $\theta$ takes values in the compact set $\mathcal{C} := \prod_{i=1}^d [\theta_{i,\min}, \theta_{i,\max}]$. From Assumption 1, for every fixed $\theta$, the Markov process $\{\mathcal{X}_n, n \geq 0\}$ is ergodic. The projection operator $\Lambda(\cdot) = (\Lambda_1(\cdot), \dots, \Lambda_d(\cdot))^\top : \mathcal{R}^d \to \mathcal{C}$. Here, $\Lambda_i(z) := \min(\max(\theta_{i,\min}, z), \theta_{i,\max})$, for $i = 1, 2, \dots, d$, projects any $z \in \mathbb{R}$ to its closest point in the interval $[\theta_{i,\min}, \theta_{i,\max}] \subset \mathbb{R}$. Define now a sequence of points $\{n_m, m \geq 0\}$, parameter update instants of Algorithm 1, as $n_0 = 0$, $n_{m+1} := \min\{j \geq n_m \mid \sum_{i=n_m+1}^j \alpha_i \geq \beta_{n_m}\}$. It is easy to see that $\{n_m, m \geq 0\}$ is a deterministically increasing sequence of points.

Our proposed algorithm makes use of two simulations governed by $\{\hat{\theta}_j^k, j \geq 0\}$, where $k = 1, 2$, and $n_m < j \leq n_{m+1}, m \geq 0$, with $\hat{\theta}_j^1 = \theta(m) + \delta\Delta(m)$, and $\hat{\theta}_j^2 = \theta(m), m \geq 0$. Further, parameter $\theta$ is defined as $\theta(m) = (\theta_1(m), \theta_2(m), \dots, \theta_d(m))^\top$. We sample the perturbation vector $\Delta$ once $\theta(m)$ is obtained and denote the same as $\Delta(m) = (\Delta_1(m), \Delta_2(m), \dots, \Delta_d(m))^\top$, where $\Delta_i(m)$, for $i = 1, \dots d$, are mutually independent, $\pm 1$-valued, symmetric Bernoulli random variables with zero-mean. Moreover, $\Delta(m), m \geq 0$ is independent of the sigma-field generated by $\{\theta(l), l \leq m, \Delta(l), l < m\}, m \geq 0$.

In the Algorithm 1, we take $\delta$ and $\Delta$ (explained previously) as input and initialize policy parameter $\theta$ as $\theta_0$. At each time instant $n$, we get value of step-sizes $\alpha_n, \beta_n$ and from these values we compute $n_m$. We get rewards $r(n)$ and $r^+(n)$ using two parallel simulations governed by policy parameter $\theta$ and perturbed policy parameter $\theta + \delta\Delta$, respectively, at time $n$. Next, we update parameter $\theta$ at each instant $n_m$ as follows:

$$\theta_i(m+1) = \Lambda_i \left[\theta_i(m) + \left(\sum_{j=n_m+1}^{n_{m+1}} \alpha_j \frac{r^+(j) - r(j)}{\delta\Delta_i(m)}\right)\right], \tag{7}$$

for $i = 1, 2, \dots, d$. Note that the $(m+1)$th update of $\theta$ happens at instant $n_{m+1}$, see line no. 4 of the algorithm. From construction, note that $\{n_m\}$ is a subsequence of $\{n\}$.

In this algorithm, the long-run average reward $J(n)$ at time instant $n$ is calculated iteratively (see line no. 13). As $\beta_n > \alpha_n, \forall n \geq 0$, $J(n)$ evolves on a faster timescale than $\theta$, and in the long run $J(n)$ converges to $J_{\pi^\theta}$ (Borkar, 2022). See (3) for $J_{\pi^\theta}$ but for the parameterized policy $\pi^\theta$, i.e., see (34) in the Appendix. We obtain the optimal value of $\theta$, that is $\theta^*$ and the total accumulated reward $J$ after the convergence of Algorithm 1. The proposed algorithm is a purely data-driven algorithm that employs two parallel simulations for the gradient estimation.

## 4.2 Safe-Actor-only Algorithm (SAOA) for Constrained RL

Similar to Algorithm 1, in Algorithm 2, Assumptions 1 and 2 continue to hold. Further, the parameterized policy is as in (6), and the framework to use SPSA is as in section 4.1. The proposed SAOA (Algorithm 2)

involves two separate recursions but requires three sets of step-sizes $\{\zeta_n, n \geq 0\}$, $\{\alpha_n, n \geq 0\}$, and $\{\beta_n, n \geq 0\}$, respectively, satisfy (31)–(33) shown in the Appendix A.

In Algorithm 2, the recursion for the Lagrange multiplier $\eta$-update is run on the slowest timescale obtained from $\zeta_n, n \geq 0$, while the recursion for the actor parameter $\theta$-update is using the timescale $\alpha_n, n \geq 0$. In Algorithm 2, The goal of the agent is to maximize the long-run average reward while maintaining the safety cost constraint. In this algorithm, we take a small positive number $\delta$, $\Delta$ (as in Algorithm 1), cost constraints $\nu_q, q = 0, \ldots, N$ as input.

---

**Algorithm 2** Safe-Actor-only Algorithm for Constrained RL

---

**Input:** scalar $\delta > 0$, sample $\Delta$ obtained from zero-mean, $\pm 1$-valued, Bernoulli distribution.
**Input:** $\nu_q > 0$, for $q = 1, \cdots, N$.
**Output:** Optimal policy
 1: Initialisation: $\theta(0) = \theta_0$, $n_0 = 0$, $m = 0$, $0 < \sigma_4 < \sigma_5 < \sigma_6 \leq 1$.
 2: Initialisation: $J(0) = \mathcal{G}_q(0) = 0, \eta_q(0) = \eta_0$.
 3: **for** $n = 0$ to $\infty$ **do**
 4:     $\zeta_n = \frac{1}{\{1+n\}^{\sigma_6}}$, $\alpha_n = \frac{1}{\{1+n\}^{\sigma_5}}$, $\beta_n = \frac{1}{\{1+n\}^{\sigma_4}}$ satisfy $\zeta_n < \alpha_n < \beta_n$
 5:     $n_{m+1} = \min\{j \geq n_m \mid \sum_{i=n_m+1}^{j} \alpha_i \geq \beta(n_m)\}$
 6:     Get next state $s'$, reward $r(n)$, cost $g_q(n)$ using current state $s$ and action $a \sim \pi^\theta$.
 7:     $h(n) = r(n) - \sum_{q=1}^{N} \eta_q(g_q(n) - \nu_q)$
 8:     Get next state $s^+$, reward $r^+(n)$, cost $g^+(n)$ using current state $s$ and action $a \sim \pi^{\theta+\delta\Delta}$.
 9:     $h^+(n) = r^+(n) - \sum_{q=1}^{N} \eta_q(g_q^+(n) - \nu_q)$
10:     **if** $n == n_{m+1}$ **then**
11:         **for** $i = 1, \ldots, d$ **do**
12:             $\theta_i(m+1) = \Lambda_i[\theta_i(m) + (\sum_{j=n_m+1}^{n_{m+1}} \alpha_j \frac{h^+(j)-h(j)}{\delta\Delta_i(m)})]$
13:         **end for**
14:         $m \leftarrow m + 1$
15:     **end if**
16:     $J(n+1) \leftarrow J(n) + \beta_n(r(n) - J(n))$
17:     $\mathcal{L}(n+1) \leftarrow \mathcal{L}(n) + \beta_n(h(n) - \mathcal{L}(n))$
18:     $\mathcal{G}_q(n+1) \leftarrow \mathcal{G}_q(n) + \beta_n(g_q(n) - \mathcal{G}_q(n))$
19:     $\eta_q(n+1) \leftarrow \hat{\Lambda}(\eta_q(n) - \zeta_n(\mathcal{G}_q(n) - \nu_q(n)))$
20:     Update the current state $s$ as the next state $s'$, i.e., $s \leftarrow s'$
21: **end for**
22: **return** Optimal policy parameter $\theta^*$.

---

We initialize the policy parameter $\theta$, average reward $J$, average cost $\mathcal{G}_q$, Lagrange parameter $\eta_q$ (see Lines 1-2 in Algorithm 2). For each time instant $n$, we get values of step-size sequences and instants $n_m$ as described in the algorithm. Now, using two parallel simulations guided by $\theta$ and $\theta + \delta\Delta$, respectively, we obtain the corresponding reward and cost at each time instant $n$. We define $h(n) = r(n) - \sum_{q=1}^{N} \eta_q(g_q(n) - \nu_q)$ and $h^+(n) = r^+(n) - \sum_{q=1}^{N} \eta_q(g_q^+(n) - \nu_q)$ corresponding to two parallel simulations. We use the values of $h(n)$ and $h^+(n)$ in the policy parameter update at instant $n_{m+1}$ (see Line no. 12). Further, Line no. 16 calculates the average reward $J$. The update rules for the Lagrangian $\mathcal{L}$, policy parameter $\theta$, estimated cost $\mathcal{G}$, and Lagrange parameters $\eta$ are as follows:

$$\mathcal{L}(n+1) = \mathcal{L}(n) + \beta_n(h(n) - \mathcal{L}(n)), \tag{8}$$

$$\theta_i(m+1) = \Lambda_i \left[ \theta_i(m) + \left( \sum_{j=n_m+1}^{n_{m+1}} \alpha_j \frac{h^+(j) - h(j)}{\delta\Delta_i(m)} \right) \right], \text{ for } i = 1, 2, \ldots, d. \tag{9}$$

$$\mathcal{G}_q(n+1) = \mathcal{G}_q(n) + \beta_n(g_q(n) - \mathcal{G}_q(n)), \tag{10}$$

$$\eta_q(n+1) = \hat{\Lambda}(\eta_q(n) - \zeta_n(\mathcal{G}_q(n) - \nu_q(n))), \text{ for } q = 1, 2, \ldots, N. \tag{11}$$

In (11), $\hat{\Lambda} : \mathbb{R} \to [0, B_\eta]$ denotes the projection of $z$, $\hat{\Lambda}(z) = \max(0, \min(z, B_\eta))$, for any $z \in \mathbb{R}$, where $B_\eta < \infty$ is a large positive constant. $\hat{\Lambda}$ ensures boundedness of Lagrange parameters $\eta_q$. Upon convergence of Algorithm 2, we obtain the optimal parameter $\theta^*$ that provides the optimal safe policy $\pi^{\theta^*}$. Further, Algorithm 2 provides the average total reward $J$, average safety cost $\mathcal{G}_q$, and the optimal value of Lagrange parameters $\eta_q$ upon convergence.

## 5 Asymptotic Convergence Analysis

We first present the asymptotic convergence results of our proposed algorithm AOA, i.e., Algorithm 1. Subsequently, we briefly sketch the changes in analysis needed for the (constrained algorithm) SAOA, i.e., Algorithm 2. The detailed proof of all Lemmas and Theorems is provided in Appendix A.1.

### 5.1 Asymptotic Analysis of Algorithm 1

**Lemma 1.** $J_{\pi^\theta}$ *is continuously differentiable in* $\theta \in \mathcal{C}$, *where* $\mathcal{C}$ *is a compact set.*

For purposes of the remaining analysis, we alternatively consider a cost minimization problem (instead of a reward maximization) wherein we set $c(j) = -r(j)$, $\forall j$. While Algorithm 1 maximizes the long-term reward $J_{\pi^\theta}$, an algorithm with the aforementioned cost structure would minimize $\bar{J}_{\pi^\theta} = -J_{\pi^\theta}$. Thus, (34) and (7) can be rewritten in terms of long-term average cost $\bar{J}_{\pi^\theta}$ and single-stage cost as follows:

$$\bar{J}_{\pi^\theta} = \lim_{n \to \infty} \frac{1}{n} \mathbb{E}[\sum_{k=0}^{n-1} c_{k+1} \mid \pi^\theta] \tag{12}$$

$$\theta_i(m+1) = \Lambda_i \left[ \theta_i(m) - \left( \sum_{j=n_m+1}^{n_{m+1}} \alpha_j \frac{c^+(j) - c(j)}{\delta \Delta_i(m)} \right) \right], \tag{13}$$

for $i = 1, 2, \ldots, d$. We use here $\bar{J}_{\pi^\theta}$ and $\bar{J}(\theta)$ interchangeably to mean the same quantity. We analyze the convergence of (13) to prove the convergence of Algorithm 1.

Let $K$ denote the set of all stationary points of the function $\bar{J}$, i.e.,

$$K = \{\theta \in \mathcal{C} \mid \bar{\Lambda}\left(-\nabla \bar{J}(\theta)\right) = 0\}, \tag{14}$$

where for any bounded, continuous, real-valued function $v(\cdot)$, $y \in \mathcal{C}$, $i = 1, \ldots, d$,

$$\bar{\Lambda}_i(v(y)) = \lim_{\eta \downarrow 0} \left( \frac{\Lambda_i(y + \eta v(y)) - y}{\eta} \right). \tag{15}$$

Also, $\bar{\Lambda}(x) = (\bar{\Lambda}_i(x_i), i = 1, \ldots, d)^T$, where $x = (x_i, i = 1, \ldots, d)^T$. The operator $\bar{\Lambda}(\cdot)$ ensures that the evolution of the ODE happens within the set $\mathcal{C}$. Further, given $\gamma > 0$, let $K^\gamma$ denote the $\gamma$-neighborhood of the set $K$, i.e.,

$$K^\gamma = \{\theta \in \mathcal{C} \mid \|\theta - \theta_0\| < \gamma, \ \theta_0 \in K\}.$$

**Theorem 1.** *Given* $\gamma > 0$, $\exists \delta_0 > 0$ *such that for any* $\delta \in (0, \delta_0]$, *the iterates* $\theta(n), n \geq 0$ *governed by Algorithm 1 converge to* $K^\gamma$ *almost surely.*

### 5.2 Asymptotic Analysis of Algorithm 2

**Lemma 2.** $\mathcal{L}_{\pi^\theta, \eta}$ *is continuously differentiable in* $\theta \in \mathcal{C}$, *where* $\mathcal{C}$ *is a compact set.*

Algorithm 2 maximizes $\mathcal{L}_{\pi^\theta,\eta}$, i.e., is equivalent to minimizing $\bar{\mathcal{L}}_{\pi^\theta,\eta}$. Let, at time instance $j$ single-stage cost $\bar{h}(j) = -h(j)$ (here if $h(j)$ is uniformly bounded, then $\bar{h}(j)$ is also uniformly bounded) and

$$\bar{\mathcal{L}}_{\pi^\theta,\eta} = \lim_{n \to \infty} \frac{1}{n} \mathbb{E}[\sum_{k=0}^{n-1} \bar{h}_{k+1} \mid \pi^\theta]. \tag{16}$$

Further, rewriting (9) we get, $\theta_i(m+1) = \Lambda_i \left[ \theta_i(m) - \left( \sum_{j=n_m+1}^{n_{m+1}} \alpha_j \frac{\bar{h}^+(j) - \bar{h}(j)}{\delta \Delta_i(m)} \right) \right],$ \tag{17}

for $i = 1, 2, \ldots, d$. In this section, we use $\bar{\mathcal{L}}_{\pi^\theta,\eta}$ and $\bar{\mathcal{L}}(\theta, \eta)$ interchangeably to mean the same quantity. We receive the following convergence of Algorithm 2 analyzing (17) and (11). Now, recall from line 4 of Algorithm 2 and (28)–(30), Appendix A, that $\eta$-update in (11) runs in the slower time-scale than $\theta$-update in (17). Thus, from the viewpoint of $\theta$-update recursion, the recursion of $\eta$-update would appear to be quasi-static (Borkar, 2022). Hence, we show the convergence of $\theta$-update in Theorem 2 for a given value of $\eta$. Further, to prove the convergence of $\eta$-update in Proposition 1, $\theta$-update recursion would appear to have converged. In the below, the definition of $K_\gamma$ is similar as $K^\gamma$ and can get by replacing function $\bar{J}$ with Lagrange function $\bar{\mathcal{L}}$.

**Theorem 2.** *Given lagrange multiplier $\eta$, $\gamma > 0$, $\exists \delta_0 > 0$ such that for any $\delta \in (0, \delta_0]$ policy parameter $\theta$, in the Algorithm 2 converges to $K_\gamma$ almost surely.*

Note that, the convergence of $\eta = (\eta_1, \ldots, \eta_N)^\top$ is shown in Proposition 1 in Appendix A.1.2.

## 6 Finite Time Analysis

We here present the finite-time sample complexity of our Algorithm 1 and Algorithm 2. The detailed proof of all Lemmas and Theorems is provided in Appendix A.2.

### 6.1 Finite Time Analysis of Algorithm 1

In this section, we discuss the finite time analysis as in (Wu et al., 2020) but for our proposed Algorithm 1. We make the required assumption as follows:

**Assumption 3.** *(Uniform ergodicity). For a fixed $\theta$, as before, let $d_{\pi^\theta}(\cdot)$ be the stationary distribution induced by the policy $\pi^\theta(s, \cdot)$ and the transition probabilities $P(\cdot \mid s, a)$. Consider a Markov chain generated by the rule $a_t \sim \pi^\theta(s_t, \cdot), s_{t+1} \sim P(\cdot \mid s_t, a_t)$. Then there exists $\vartheta > 0$ and $\rho \in (0, 1)$ such that:*

$$d_{TV}(P(s_\iota \in \cdot \mid s_0 = s), d_{\pi^\theta}(\cdot)) \le \vartheta \rho^\iota, \forall \iota \ge 0, \forall s \in S \tag{18}$$

In the above, $d_{TV}(O, Q)$ is defined as $d_{TV}(O, Q) = 0.5 * \int_{\mathcal{Y}} | O(dy) - Q(dy) |$ and called total variation distance of two probability measures $O$ and $Q$. Further, we define an integer that depends on the learning rates in Algorithm 1, as follows:

$$\iota_m \triangleq \min\{m \ge 0 \mid \vartheta \rho^{m-1} \le \min\{ \sum_{i=n_{m-1}+1}^{n_m} \alpha_i, \beta(m)\}\}, \tag{19}$$

where $\vartheta, \rho$ are defined in Assumption 3. By definition, in (19), $\iota_m$ is the mixing time of an ergodic Markov chain and is used to control the Markovian noise encountered during the training process. Now, we rewrite

and analyze the recursion (13).

$$
\begin{aligned}
\theta_i(m+1) =& \Lambda_i(\theta_i(m) - \beta(m)\frac{\bar{J}(\theta(m)+\delta\Delta(m)) - \bar{J}(\theta(m))}{\delta\Delta_i(m)} - \\
& \beta(m)[\frac{\sum_{j=n_m+1}^{n_{m+1}} \alpha_j\left(\frac{c^+(j)-c(j)}{\delta\Delta_i(m)}\right)}{\beta(m)} - \frac{\bar{J}(\theta(m)+\delta\Delta(m)) - \bar{J}(\theta(m))}{\delta\Delta_i(m)}]) \\
=& \Lambda_i(\theta_i(m) - \beta(m)\frac{\bar{J}(\theta(m)+\delta\Delta(m)) - \bar{J}(\theta(m))}{\delta\Delta_i(m)}) - \\
& \frac{\beta(m)}{\delta\Delta_i(m)}\left[\frac{\sum_{j=n_m+1}^{n_{m+1}} \alpha_j\left(c^+(j)-c(j)\right)}{\beta(m)} - (\bar{J}(\theta(m)+\delta\Delta(m)) - \bar{J}(\theta(m)))\right]
\end{aligned}
\tag{20}
$$

$$
\therefore \theta_i(m+1) = \Lambda_i[\theta_i(m) - \beta(m)\frac{\bar{J}(\theta(m)+\delta\Delta(m)) - \bar{J}(\theta(m))}{\delta\Delta_i(m)} - \beta(m)\mathcal{N}(\theta_i(m))]
\tag{21}
$$

$$
= \Lambda_i\left[\theta_i(m) - \beta(m)\left(\hat{\nabla}\bar{J}(\theta(m)) + \mathcal{N}(\theta_i(m))\right)\right],
\tag{22}
$$

where $\mathcal{N}(\theta_i(m)) = \frac{1}{\delta\Delta_i(m)}[\frac{\sum_{j=n_m+1}^{n_{m+1}} \alpha_j\left(c^+(j)-c(j)\right)}{\beta(m)} - (\bar{J}(\theta(m)+\delta\Delta(m)) - \bar{J}(\theta(m)))]$.

We now analyze the recursion (22) considering $\bar{J}(\cdot)$ is a non-convex function. First, we introduce the required Lemmas, and then discuss the main theorem. Now, let, $\mathbb{E}_m$ is shorthand for $\mathbb{E}\left(\cdot \mid \mathcal{F}_m\right)$, where $\mathcal{F}_m$ be the sigma-field generated by $\{\theta(l), l < m\}, m \geq 0$.

**Lemma 3.** *The faster and slower step sizes $\beta_n = \frac{1}{\{1+n\}^{\sigma''}}$, $\alpha_n = \frac{1}{\{1+n\}^{\sigma'''}}$, where $0 < \sigma'' < \sigma'''$, follows $0 \leq \frac{\sum_{j=n_m+1}^{n_{m+1}} \alpha_j}{\beta_{m_n}} \leq c''$, assuming $0 \leq n_{m+1} - n_m \leq c''\{n_m\}^{\sigma'}$, $c'' > 0$ where $0 < \sigma' + \sigma'' < \sigma'''$. Thus, $\max_m \frac{\sum_{j=n_m+1}^{n_{m+1}} \alpha_j}{\beta(m)}$ is bounded.*

**Lemma 4.** *$\mathbb{E}_m[\|\mathcal{N}(\theta(m))\|] \leq \frac{B_1\beta_m}{\delta}$ and $\mathbb{E}_m[\|\mathcal{N}(\theta(m)\|^2] \leq \frac{B_4\beta_m^2}{\delta^2}$ for some constant $B_1, B_4 > 0$.*

**Lemma 5.** *There exists a constant $B > 0$ such that $\|\nabla\bar{J}(\theta)\|_1 \leq B, \forall\theta \in \mathbb{R}^d$.*

**Lemma 6.** *The gradient estimate $\hat{\nabla}\bar{J}(\theta_k)$ satisfies the following inequalities for all $k \geq 1$ :*

$$
\left\|\mathbb{E}_k\left[\hat{\nabla}\bar{J}(\theta_k)\right] - \nabla\bar{J}(\theta_k)\right\|_\infty \leq c_1\delta \qquad (23) \qquad \mathbb{E}_k\left[\left\|\hat{\nabla}\bar{J}(\theta_k)\right\|^2\right] \leq \left\|\mathbb{E}_k\left[\hat{\nabla}\bar{J}(\theta_k)\right]\right\|^2 + \frac{c_2}{\delta^2} \qquad (24)
$$

In the above, $\mathbb{E}_k$ is shorthand for $\mathbb{E}\left(\cdot \mid \mathcal{F}_k\right)$, with sigma-field $\mathcal{F}_k$ and $c_1, c_2$ are some positive constants. The significance and required assumptions for Lemma 3 –Lemma 6 are summarized at Remark 1.

**Remark 1.** *1. Lemma 3 confirms the boundedness of $\frac{\sum_{j=n_m+1}^{n_{m+1}} \alpha_j}{\beta(m)}$ at finite time instant $m$. Assumption $0 \leq n_{m+1} - n_m \leq c''\{n_m\}^{\sigma'}$, $c'' > 0$ where $0 < \sigma' + \sigma'' < \sigma'''$ is used in this lemma.*
*2. Lemma 4 confirms boundedness of the noise term, and the square of the noise term $\mathcal{N}$ at finite time instant $m$. Assumption 3 and Lemma 3 are used in this Lemma.*
*3. Lemma 5 confirms that for all $\theta$, gradient of the cost function $\bar{J}(\theta)$ is bounded. In this lemma, it is assumed that gradient of $\bar{J}(\cdot)$, i.e., $\nabla\bar{J}(\cdot)$ is estimated by $\hat{\nabla}\bar{J}(\cdot)$ and this is true, see equations (48) and (49) in Appendix A.2.1.*
*4. Lemma 6 confirms that the difference between the expected value of the estimated gradient and the exact gradient is bounded. No assumption is made for this lemma.*

**Remark 2.** *As a consequence of Lemma 5 it follows that $\|\nabla\bar{J}(\theta') - \nabla\bar{J}(\theta'')\|_1 \leq B \leq L\|\theta' - \theta''\|_1, \forall \theta', \theta'' \in \mathbb{R}^d$. Thus, $\bar{J}$ is L-smooth (as in Definition 2 shown in the Appendix A.2), where $L = B$ and $L > 0$.*

**Definition 1.** *Iteration complexity: For a given $\epsilon > 0$, the iteration complexity of an algorithm is the number of iterations of the algorithm before finding an $\epsilon$-stationary point for a non-convex objective function.*

**Theorem 3.** *Suppose the objective function $\bar{J}$ is L-smooth (as in (A. & Bhatnagar, 2024; Wu et al., 2020)), and Lemma 5 - 6 hold. Suppose that the recursion (22) is run with the stepsize $\beta_k$ for each $k = \iota_m, \ldots, m$, where $\beta_k = \min\left\{\frac{1}{L}, \frac{1}{\{1+k\}^{\sigma''}}\right\}$. Then an order $\mathcal{O}(\epsilon^{-2})$ iterations of the Algorithm 1 are enough to find a point $\theta_k$ that satisfies $\min_{0 \leq k \leq m} \mathbb{E}\left\|\nabla\bar{J}(\theta_k)\right\|^2 \leq \epsilon$ when $\sigma''' = 1, \sigma'' = 1/2$.*

**Proof sketch of Theorem 3:** Since $\bar{J}$ is $L$-smooth, and using Lemma 4-6 and expanding terms we get

$$\bar{J}(\theta_{k+1}) \leq \bar{J}(\theta_k) + \left\langle\nabla\bar{J}(\theta_k), \theta_{k+1} - \theta_k\right\rangle + \frac{L}{2}\left\|\theta_{k+1} - \theta_k\right\|^2$$

$$\vdots$$

$$\leq \bar{J}(\theta_k) - \left(\beta_k - \frac{L}{2}\beta_k^2\right)\left\|\nabla\bar{J}(\theta_k)\right\|^2 + c_1\delta B\left(\beta_k + L\beta_k^2\right) - \frac{BB_1\beta_k^2}{\delta} + \frac{L}{2}\beta_k^2\left[dc_1^2\delta^2 + \frac{c_2}{\delta^2}\right] + \frac{L}{2\delta^2}B_4\beta_k^4$$

Now, we rearrange terms, sum up the inequality for $k = \iota_m$ to $m$, take expectations, divide by ($1 + m - \iota_m$) both sides, and get,

$$\frac{1}{1 + m - \iota_m}\sum_{k=\iota_m}^{m}\mathbb{E}_k\left\|\nabla\bar{J}(\theta_k)\right\|^2$$

$$\leq \frac{2}{1 + m - \iota_m}\sum_{k=\iota_m}^{m}\frac{\left(\mathbb{E}_k\bar{J}(\theta_k) - \mathbb{E}_k\bar{J}(\theta_{k+1})\right)}{\beta_k(2 - L\beta_k)} + \frac{2}{1 + m - \iota_m}\sum_{k=\iota_m}^{m}c_1\delta B\left(\frac{1 + L\beta_k}{2 - L\beta_k}\right)$$

$$+ \frac{L}{1 + m - \iota_m}\sum_{k=\iota_m}^{m}\frac{\beta_k}{(2 - L\beta_k)}\left[dc_1^2\delta^2 + \frac{c_2}{\delta^2}\right] + \frac{2}{1 + m - \iota_m}\sum_{k=\iota_m}^{m}\frac{\beta_k}{(2 - L\beta_k)}\left[\frac{LB_4}{2\delta^2}\beta_k^2 - \frac{BB_1}{\delta}\right]$$

Now, upon bounding and simplifying the above right-hand side terms, we get the desired results (for detailed analysis, check Appendix A.2.1).

**Remark 3.** *Comparative analysis: The finite-time complexity $T = \mathcal{O}(\epsilon^{-2})$ of AOA (i.e., Algorithm 1) is $\mathcal{O}(\epsilon^{-2})$, $\mathcal{O}(\epsilon^{-2})$, and $\tilde{\mathcal{O}}(\epsilon^{-0.5})$ times better than (Kumar & et al., 2024),(Qiu et al., 2019), and (Wu et al., 2020), respectively, (see Table 1). Further, the notation $\tilde{\mathcal{O}}$, used in the literature, hides logarithm terms. In our case, no logarithm term is multiplied in the sample complexity as we use the $\mathcal{O}$ notation. Further, as presented in the Table 1, the finite-time complexity of algorithms in (Yujia & Sidford, 2021; Zihan & Xie, 2023; Daniil & Gasnikov, 2022) not only depends on the term $\epsilon$ but also depends on the size of the state-action space, and has linear or polynomial growth of sample complexity with the cardinality of the state and action space, unlike our algorithms. Clearly, our algorithm is better than these algorithms in terms of sample complexity. The iteration complexity in (Julien & Marek, 2023) is also in terms of the cardinality of the state-action space, unlike our algorithm. Further, our AOA algorithm achieves the sample complexity as good as in (Li et al., 2024) while having deterministic update instants, unlike (Li et al., 2024). Furthermore, our AOA not only provides better sample complexity than majority of the considered existing algorithms but also remains almost surely stable and convergent (see Theorem 1), whereas almost sure convergence is not guaranteed in (Li et al., 2024).*

## 6.2 Finite Time Analysis of Algorithm 2

In this section, we discuss the finite time analysis as in (Panda & Bhatnagar, 2024) but for our proposed Algorithm 2. Here, we consider the step sizes $\beta_n = \frac{1}{\{1+n\}^{\sigma_4}}$, $\alpha_n = \frac{1}{\{1+n\}^{\sigma_5}}$, and $\zeta_n = \frac{1}{\{1+n\}^{\sigma_6}}$ where $0 <$

---

[1] ✗, ✓ , respectively, denote not available and available.

[2] The mixing of the probability transition matrix of all policies is at most $t_{mix}$

[3] $sp(h^*)$ the span of the optimal bias function

Table 1: Finite time complexity of different algorithms [1].

| Algorithm | FT Complexity ($T$) | Asymptotic Analysis |
|---|---|---|
| discounted reward AC (Kumar & *et al.*, 2024) | $\mathcal{O}(\epsilon^{-4})$ | ✗ |
| average (avg.) reward AC (Qiu et al., 2019) | $\tilde{\mathcal{O}}(\epsilon^{-4})$ | ✗ |
| avg. reward AC (Wu et al., 2020) | $\tilde{\mathcal{O}}(\epsilon^{-2.5})$ | ✗ |
| avg. reward CAC (Panda & Bhatnagar, 2024) | $\tilde{\mathcal{O}}(\epsilon^{-2.5})$ | ✗ |
| discounted reward AC (Mandal et al., 2024) | ✗ | ✓ |
| avg. reward AC (Bhatnagar et al., 2009) | ✗ | ✓ |
| avg. reward CAC (Bhatnagar & Lakshmanan, 2012) | ✗ | ✓ |
| average-reward MDPs (Yujia & Sidford, 2021)[2] | $\tilde{\mathcal{O}}(\lvert \mathcal{S} \rvert \mathcal{A} t_{mix}\epsilon^{-3})$ | ✗ |
| average-reward MDPs (Zihan & Xie, 2023)[3] | $\tilde{\mathcal{O}}(\mathcal{S}\mathcal{A}sp^2(h^*)\epsilon^{-2} + \mathcal{S}^2\mathcal{A}sp(h^*)\epsilon^{-1})$ | ✗ |
| average-reward MDPs (Daniil & Gasnikov, 2022) | $\tilde{\mathcal{O}}(t_{mix}^2 \lvert \mathcal{S} \rvert \epsilon^{-2})$ | ✗ |
| average-reward MDPs (Li et al., 2024) | $\mathcal{O}(\epsilon^{-2})$ | ✗ |
| **AOA (Proposed)** | $\mathcal{O}(\epsilon^{-2})$ | ✓ |
| **SAOA (Proposed)** | $\mathcal{O}(\epsilon^{-2})$ | ✓ |

$\sigma_4 < \sigma_5 < \sigma_6 \leq 1$. Further, let $0 \leq r \leq B_r$, $0 \leq g_q \leq B_g$, $0 \leq \nu_q \leq B_\nu$, $0 \leq \eta_q \leq B_\eta, \forall q = 1, \ldots, N$, where $B_r$, $B_g$, $B_\nu$, $B_\eta < \infty$ are some large positive constants. Thus, $0 \leq h \leq B_h$, where $B_h = B_r + N B_\eta(B_g + B_\nu)$ and $\bar{h}$ also upper bounded by $B_h$, and $B_h < \infty$.

As in the Algorithm 1, here also we consider the Assumption 3 i.e., Uniform ergodicity. Now, we define mixing time for Algorithm 2 as follows:

$$\iota_m \triangleq \min\{m \geq 0 \mid \vartheta\rho^{m-1} \leq \min\{\zeta(m), \sum_{i=n_{m-1}+1}^{n_m} \alpha_i, \beta(m)\}\}, \tag{25}$$

where $\vartheta, \rho$ are defined in Assumption 3. By definition, in (25), $\iota_m$ is the mixing time of an ergodic Markov chain and is used to control the Markovian noise encountered during the training process. We now rewrite and analyze the recursion (17) by following the simar steps as in (20) and get

$$\theta_i(m+1) = \Lambda_i[\theta_i(m) - \beta(m)\frac{\bar{\mathcal{L}}(\theta(m) + \delta\Delta(m), \eta(m)) - \bar{\mathcal{L}}(\theta(m), \eta(m))}{\delta\Delta_i(m)} - \beta(m)\mathcal{N}_1(\theta_i(m), \eta(m))] \tag{26}$$

$$= \Lambda_i \left[\theta_i(m) - \beta(m)\left(\hat{\nabla}\bar{\mathcal{L}}(\theta(m), \eta(m)) + \mathcal{N}_1(\theta_i(m), \eta(m))\right)\right], \tag{27}$$

where $\mathcal{N}_1(\theta_i(m), \eta(m)) = \frac{1}{\delta\Delta_i(m)}[\frac{\sum_{j=n_m+1}^{n_{m+1}} \alpha_j(\bar{h}^+(j) - \bar{h}(j))}{\beta(m)} - (\bar{\mathcal{L}}(\theta(m) + \delta\Delta(m), \eta(m)) - \bar{\mathcal{L}}(\theta(m), \eta(m)))]$.

We now analyze the recursion (27) considering $\bar{\mathcal{L}}(\cdot)$ is a non-convex function. All required lemmas, Lemma 7 to Lemma 10 are presented and proved in the Appendix A.2.2. Now, let, $\mathbb{E}_m$ is shorthand for $\mathbb{E}(\cdot \mid \mathcal{F}_m)$, where $\mathcal{F}_m$ be the sigma-field generated by $\{\theta(l), l < m\}, m \geq 0$. Note that, $\bar{\mathcal{L}}$ is $L$-smooth due to Lemma 9, as in Remark 2.

**Theorem 4.** *Suppose the objective function $\bar{\mathcal{L}}$ is L-smooth (as in (A. & Bhatnagar, 2024)), and Lemma 9 - Lemma 10 hold. Suppose that the recursion (27) is run with the stepsize $\beta_k$ for each $k = \iota_m, \ldots, m$, where $\beta_k = \min\left\{\frac{1}{L}, \frac{1}{\{1+k\}^{\sigma_4}}\right\}$. Then an order $\mathcal{O}(\epsilon^{-2})$ iterations of the Algorithm 2 are enough to find a point $\theta_k$ that satisfies $\min_{0 \leq k \leq m} \mathbb{E}\left\|\nabla\bar{\mathcal{L}}(\theta_k, \eta(k))\right\|^2 \leq \epsilon$ when $\sigma_6 = 1$, $\sigma_5 = 0.99, \sigma_4 = 0.49$.*

**Remark 4.** *Comparative analysis: The similar arguments follows as in the remark 3. The finite-time complexity $T = \mathcal{O}(\epsilon^{-2})$ of our SAOA (i.e., Algorithm 2) as demonstrated in Theorem 4, and presented in the Table 1, is better than majority of considered existing algorithms, even if this algorithm is for the constrained*

*RL settings. Note that our SAOA algorithm obtained sample complexity as good as in (Li et al., 2024), while having deterministic update instants, applicable for constrained RL, remains almost surely stable and convergent (see Theorem 2), unlike (Li et al., 2024).*

## 7    Experiments and Results

In this section, we demonstrate the performance of the proposed algorithms AOA (i.e., Algorithm 1) and SAOA (i.e., Algorithm 2) on standard RL environments. Here, we consider different Grid-world (GW) environments (for continuing tasks) such as having size $10 \times 10$, $50 \times 50$, and $100 \times 100$ to ensure the scalability of our algorithms. We perform 1,00,000 training iterations (i.e., the value of $n$ in the algorithms, alternatively, the number of function measurements) to ensure the convergence and stability of the algorithms. We further employ 'CartPole' and 'Acrobot', widely used RL environments in the recent RL literature (Wang et al., 2023; Muppidi et al., 2024), available in the OpenAI gym (Brockman et al., 2016). We perform 10000 training iterations, i.e., total episodes are 10000 (for episodic tasks), and the length of each episode is $n$, which is random. In all our experiments, linear softmax policies are implemented as shown in eq. (6). In all experiments of Algorithm 1, step sizes $\alpha_n$, and $\beta_n$ are $\frac{1}{\{1+n\}}$, and $\frac{1}{\{1+n\}^{0.66}}$, respectively. In all experiments of Algorithm 2, step sizes $\zeta_n$, $\alpha_n$, and $\beta_n$ are $\frac{1}{\{1+n\}}$, $\frac{1}{\{1+n\}^{0.95}}$, and $\frac{1}{\{1+n\}^{0.66}}$, respectively. The value of $\delta$ is set to 0.6 in all experiments.

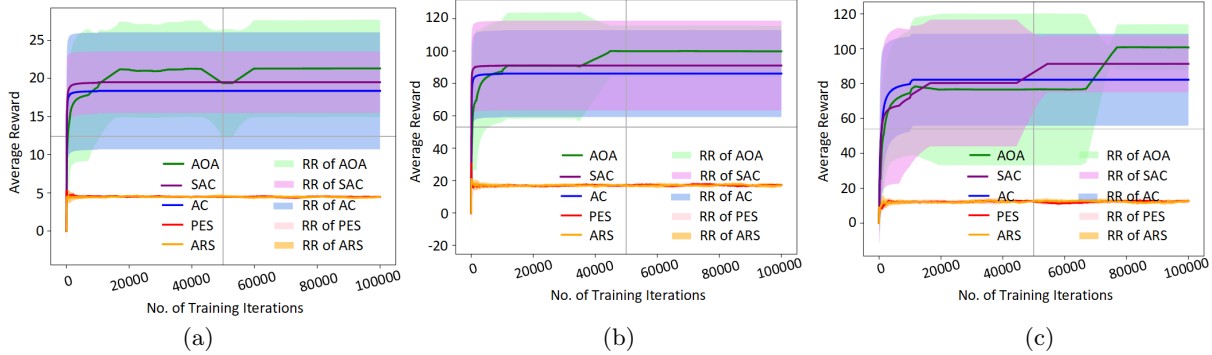

Figure 1: Comp. with AOA: Avg. reward on (a) $10 \times 10$ (b) $50 \times 50$ (c) $100 \times 100$ GW w.r.t. train. iterations.

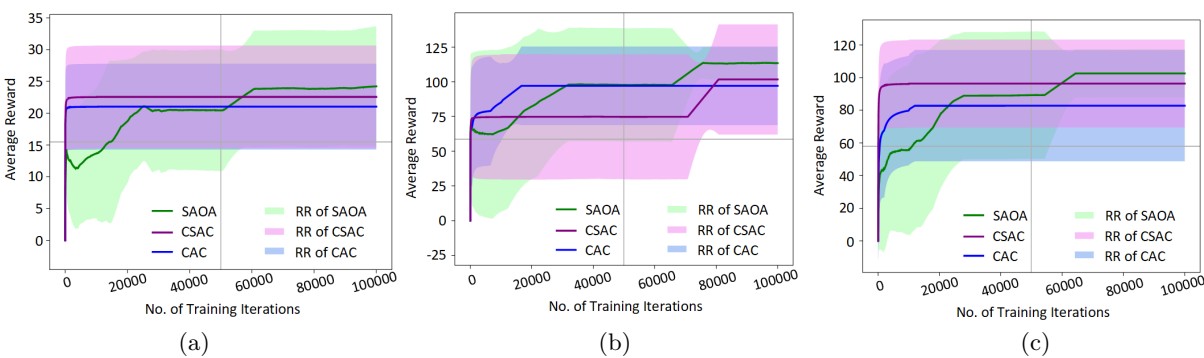

Figure 2: Comp. with SAOA: Avg. reward on (a) $10 \times 10$ (b) $50 \times 50$ (c) $100 \times 100$ GW w.r.t. iterations.

**Performance of AOA on GW:** To analyze the performance and convergence (conv.) of our AOA, we study the evolution of average (avg.) reward with respect to (w.r.t.) training iterations. We generate ten different random seeds to observe the training results of ten different independent runs. The avg. reward and standard deviation (std.) of rewards obtained at each time instant in ten different runs are calculated and plotted, where the $x$-axis presents the number (no.) of training iterations, and the $y$-axis presents the avg. total reward obtained so far. 'RR' in each plot represents the reward range, i.e., std. of rewards. The performance of the algorithm during training is presented in Figure 1(a), Figure 1(b), Figure 1(c), respectively, for three different environments with the cardinality of state and action space, i.e., $(\mid S \mid, \mid \mathcal{A} \mid)$ are $(100, 5)$, $(2500, 5)$

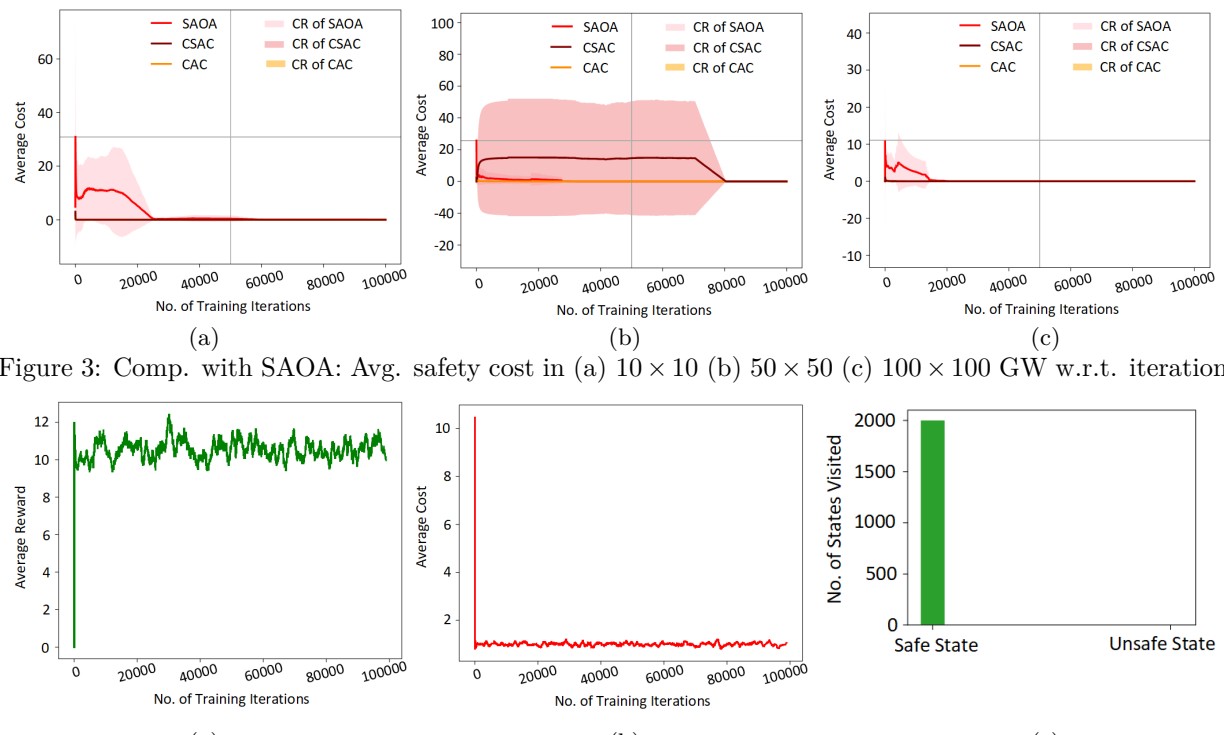

Figure 3: Comp. with SAOA: Avg. safety cost in (a) $10 \times 10$ (b) $50 \times 50$ (c) $100 \times 100$ GW w.r.t. iterations.

Figure 4: Conv. result: (a) Avg. reward (b) Avg. Safety Cost (c) No. of visit in Safe and Unsafe states.

and $(10, 000, 5)$, respectively. In each environment, some states contain positive rewards, and all other states contain reward zero, the agent's goal is to maximize the long-term reward, i.e., to maximize the visit of the maximum reward-containing state.

**Empirical comparative analysis:** We consider popular algorithms such as standard Actor-Critic (AC) (Bhatnagar et al., 2009) and Soft Actor-Critic (SAC) (Haarnoja et al., 2018), Parallelized Evolution Strategies (PES) (Salimans et al., 2017), and Augmented Random Search (ARS) (Mania et al., 2018) and perform additional experiments on the same environmental settings as in proposed AOA. From our experiments, we observe the following result:
1. We observe that the avg. total reward converges for all three considered environments.
2. Figure 1(a), Figure 1(b), Figure 1(c) demonstrate that our proposed AOA outperforms the considered algorithms AC (Bhatnagar et al., 2009), SAC (Haarnoja et al., 2018), PES (Salimans et al., 2017), ARS (Mania et al., 2018) by achieving a highest avg. total reward while training in all considered environments.
3. The converged results of PES and ARS algorithms are very lower than (i.e., not comparable to) our proposed algorithm, and hence in columns $2 - 4$ of Table 2, we present converged mean and std. of rewards of AC, SAC with our proposed AOA (see column 3). The numerical values show the highest reward of our proposed algorithm while achieving lower std. (majority of the cases).
4. Columns $2 - 6$ of Table 3 shows that our AOA achieved $15.23\% - 99.31\%$ performance improvement (alternatively computational cost reduction) in terms of computational time compared to considered algorithms.

**Performance of SAOA on GW:** We now study the evolution of avg. reward and safety cost w.r.t. training iterations to analyze the empirical performance and convergence of the proposed SAOA. For the SAOA, we also consider the state and action spaces with the same cardinality as discussed above for the AOA in three different GWs. However, here we have unsafe states having unwanted costs with the reward-giving states, in the state space and need to avoid those unsafe states while maximizing the avg. reward.

**Empirical comparative analysis:** We consider SOTA algorithms constrained Actor-Critic (CAC) (Bhatnagar & Lakshmanan, 2012) and constrained Soft Actor-Critic (CSAC) (Haarnoja et al., 2018) for the comparative analysis. The avg. reward and std. of rewards obtained at each time instant of ten different independent runs of the algorithm are calculated and plotted in Figure 2(a), Figure 2(b), Figure 2(c), respectively, for considered three different GW settings where the $x$ and $y$ axis are as in Figure 1. Further, in a

Table 2: Performance (mean ± std. of rewards) across 10 independent runs (using 10 random seeds).

| Environment | AC | SAC | Proposed AOA | CAC | CSAC | Proposed SAOA |
|---|---|---|---|---|---|---|
| 10 × 10 GW | 18.37 ± 7.67 | 19.50 ± 4.03 | **21.31 ± 6.38** | 21.02 ± 6.75 | 22.55 ± 8.09 | **24.22 ± 9.47** |
| 50 × 50 GW | 86.08 ±26.85 | 91.08 ± 27.71 | **99.84 ±15.77** | 97.06 ±28.28 | 101.67 ± 39.66 | **113.40 ±12.42** |
| 100 × 100 GW | 82.21 ± 26.32 | 91.31 ± 16.35 | **100.71 ± 13.19** | 82.81 ± 34.09 | 96.38 ± 26.90 | **102.55 ± 14.76** |

Table 3: Computation time (in seconds) for $n = 1,00,000$ iterations.

| No. of Itr. | AC | SAC | PES | ARS | Proposed AOA | CAC | CSAC | Proposed SAOA |
|---|---|---|---|---|---|---|---|---|
| 1,00,000 | 49.52 | 53.73 | 776.91 | 6126.43 | **41.98** | 62.29 | 67.54 | **61.65** |

similar manner, the avg. cost is calculated and plotted in Figure 3(a), Figure 3(b), Figure 3(c), respectively. In each figure, the $x$-axis presents as in Figure 3, and the $y$-axis presents the obtained avg. total cost. 'CR' in each plot represents the cost range, i.e., std. of costs. From the training of SAOA, we observe the following:
**1.** Each part of Figure 2 and Figure 3 show that the avg. total reward and avg. total safety cost, respectively, converge for all considered experimental setups.
**2.** For all three environments, the cost constraint is satisfied for different values of $\nu$. Here we show that for $\nu = 0.01$ the safety cost satisfies the constraint as the converged avg. cost is 0.
**3.** The columns $5 - 7$ of Table 2 and columns $7 - 9$ of Table 3 demonstrate that our proposed SAOA outperforms the considered algorithms CAC (Bhatnagar & Lakshmanan, 2012) and CSAC (Haarnoja et al., 2018) by achieving a highest avg. total reward and the least computation time in the constrained setup in all three experimental environments.

Further, we train our SAOA in $\mid S \mid= 100, \mid \mathcal{A} \mid= 4$ setup and present the avg. total rewards, avg. total cost, and after convergence, the no. of visits of "safe" and "unsafe" states. We can observe from Figure 4(a), Figure 4(b), respectively, that avg. total rewards converge, avg. total costs converge and satisfy the cost constraint $\nu = 1$. Figure 4(c) shows that the no. of visit to "unsafe" sate in last 2000 iterations i.e., after convergence is zero.

**Performance on CartPole and Acrobot environment:** Figure 5 and Table 4 demonstrate performance evaluation during training and after convergence in the 'CartPole' and 'Acrobot' environments. For the performance comparison, we consider two popular policy gradient (PG) algorithms, Monte-Carlo PG (MCPG) and AC. However, since the results obtained using MCPG are much lower than our proposed algorithm (see Table 4), in the training plot we present our algorithm and AC. Parts (a) and (b) of Figure 5 show the training evolution of our AOA and existing AC algorithms. Table 4 (as a representative of our proposed algorithms) shows the average reward after convergence and overall computational time (for 10000 episodes) performance (in seconds). Our AOA obtained approx. 3 and 4 times performance improvement in terms of computation time compared to existing algorithms in the CartPole and Acrobot environments. Part (c) of Figure 5 (as a representative of both environments) shows the average cost during training on Cartpole

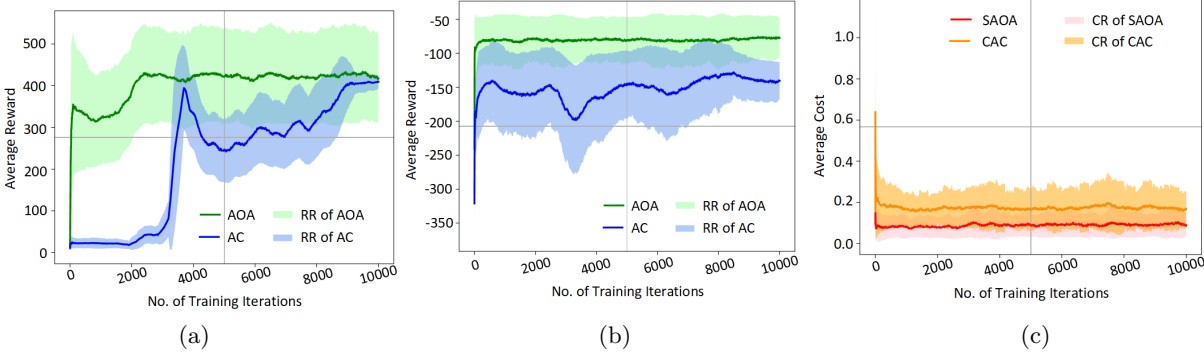

Figure 5: Performance in terms of average reward or average cost on different environments (a) AOA, AC on CartPole (b) AOA, AC on Acrobot (c) SAOA, CAC on CartPole, w.r.t. train. iterations.

Table 4: Performance after convergence on CartPole and Acrobot environments.

| Environments | Average Reward | | | Computation Time $\times 10^2$ (Sec.) | | |
|---|---|---|---|---|---|---|
| | MCPG | AC | **AOA** | MCPG | AC | **AOA** |
| CartPole-v1 | 232.6 | 409.4 | **417.8** | 0.3740 | 0.3313 | **0.1016** |
| Acrobot-v1 | -193.7 | -140.5 | **-77.2** | 8.8948 | 10.9900 | **2.8589** |

of SAOA and existing CAC. After convergence, the obtained average cost is 0.09 and 0.17, respectively, for SAOA and CAC, which shows that our SAOA is better than CAC and satisfies the cost constraint $\nu = 0.5$. Note that in the CartPole environment for the constrained RL algorithms, the cost can be calculated by penalizing the large pole angle, the fast cart velocity, or both (considered in our settings). Figure 5 and Table 4 present that our algorithms outperform the considered algorithms in terms of average reward /average cost and computation time. Further, it is confirmed that the performance of our algorithms is generalizable across environments.

## 8 Conclusions

We propose Actor-only (AOA) and Safe-Actor-only (SAOA) reinforcement learning algorithms, where we introduce a procedure to determine the policy update instants. Our proposed algorithm eliminates the uncertainty of policy update that exists in the regular Monte-Carlo PG methods. Our AOA incorporates two timescales - one each for getting policy update instants and updating policy parameters, respectively. Our SAOA incorporates three timescales - two timescales as in AOA and a third timescale to update the Lagrange multiplier. We provide asymptotic convergence as well as finite-time analysis of our proposed algorithms, and empirically demonstrate the convergence of the proposed algorithms. The finite-time analysis of our algorithms demonstrates that our algorithms outperform the majority of the considered algorithms in the literature, even though our algorithms are two-timescale and three-timescale algorithms. Further, our experimental results across different RL environments show a better performance of our algorithms both in terms of computational time and average total reward than the considered algorithms in the literature.

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

# A    Appendix

In this appendix, we discuss and prove the lemma and theorem from Section 5 and 6.

**Notations:**

- $\sigma'', \sigma''', \sigma_4, \sigma_4, \sigma_5, \sigma_6$ are positive constants and take values as defined in the algorithms and theorems.

- $\sigma', \sigma_3$ are positive constants and are employed in Lemma 3 and Lemma 7, respectively.

**Step-size sequences criterion:**
Step-size sequences in the Algorithm 1 satisfy the following criterion.

$$\sum_n \alpha_n = \sum_n \beta_n = \infty; \ \alpha_n, \beta_n > 0, \forall n \geq 0, \tag{28}$$

$$\sum_n (\alpha_n{}^2 + \beta_n{}^2) < \infty; \ \lim_{n \to \infty} \frac{\alpha_n}{\beta_n} = 0. \tag{29}$$

$$\lim_{n \to \infty} \frac{\alpha_{n+1}}{\alpha_n} = 1; \ \lim_{n \to \infty} \frac{\beta_{n+1}}{\beta_n} = 1. \tag{30}$$

Step-size sequences in the Algorithm 2 satisfy the following.

$$\sum_n \zeta_n = \sum_n \alpha_n = \sum_n \beta_n = \infty; \ \zeta_n, \alpha_n, \beta_n > 0, \forall n \geq 0, \tag{31}$$

$$\sum_n (\zeta_n{}^2 + \alpha_n{}^2 + \beta_n{}^2) < \infty; \ \lim_{n \to \infty} \frac{\zeta_n}{\alpha_n} = 0. \ \lim_{n \to \infty} \frac{\alpha_n}{\beta_n} = 0. \tag{32}$$

$$\lim_{n \to \infty} \frac{\alpha_{n+1}}{\alpha_n} = 1; \ \lim_{n \to \infty} \frac{\beta_{n+1}}{\beta_n} = 1. \tag{33}$$

## A.1    Asymptotic Convergence Analysis

### A.1.1    Asymptotic Analysis of Algorithm 1

**Proof of Lemma 1:**

*Proof.* Recall that

$$J_{\pi^\theta} \ = \ \lim_{n \to \infty} \frac{1}{n} \mathbb{E}[\sum_{k=0}^{n-1} r_{k+1} \mid \pi^\theta] = \sum_{s \in \mathcal{S}} d_{\pi^\theta}(s) \sum_{a \in \mathcal{A}} \pi^\theta(s, a) r(s, a), \tag{34}$$

where $d_{\pi^\theta} = (d_{\pi^\theta}(s), s \in \mathcal{S})$ is the stationary probability distribution of the ergodic Markov process $\{\mathcal{X}_n, n \geq 0\}$ (see Assumption 1). Note that the expected value of single-stage reward $r(s, a)$ is uniformly bounded. Also, from Assumption 2, $\pi^\theta(s, a)$ is continuously differentiable with respect to $\theta$. In particular, as noted previously, from the form of $\pi^\theta(s, a)$ that we use, see (6), $\pi^\theta(s, a)$ is continuously differentiable with respect

to $\theta$. We now have to verify that the steady state distribution $d_{\pi^\theta} = (d_{\pi^\theta}(s), s \in S)$ is continuously differentiable.

For simplicity, let $P(\theta)$ denote the transition probability matrix of the Markov chain under policy $\pi^\theta$. In other words, $P(\theta) = [[p_{i,j}(\theta)]]_{i,j\in S}$ where $p_{i,j}(\theta) = \sum_{a\in\mathcal{A}} \pi^\theta(i,a)P(j|i,a)$. Also, let $Z(\theta) = [I - P(\theta) + P^\infty(\theta)]^{-1}$, where $I$ is the identity matrix, $P^\infty(\theta) = \lim_{m\to\infty} \frac{1}{m}\sum_{k=1}^{m} P^k(\theta)$ is the time averaged transition probability matrix, with $P^k(\theta)$ being the $k$-step transition probability matrix. Since the state-valued process is ergodic Markov for any $\theta$ (cf. Assumption 1), it follows that $P_{ij}^\infty(\theta) = d_{\pi^\theta}(j), \forall i,j \in S$. From Assumption 2, $\nabla\pi^\theta$ exists and is in fact uniformly bounded over all $\theta \in \mathcal{C} \subset \mathbb{R}^d$, a compact set. Thus, $\nabla P(\theta)$ exists as well and is also uniformly bounded. Now, recall from Section 3, that our MDP has a finite state and action space. Thus, for any policy $\pi^\theta$, the resulting Markov chain $\{\mathcal{X}_n\}$ has a finite state space. It then follows from Theorem 2 of (Schweitzer, 1968) that $d_{\pi^\theta}$ is continuously differentiable and in fact,

$$\nabla d_{\pi^\theta} = d_{\pi^\theta}\nabla P(\theta)Z(\theta).$$

It now follows from (34) that $J_{\pi^\theta}$ is continuously differentiable in $\theta \in \mathcal{C}$. □

**Proof of Theorem 1**:

*Proof.* Note that the recursion (13) can be rewritten as:

$$\theta_i(m+1) = \Lambda_i\left(\theta_i(m) - \beta(m)\frac{\sum_{j=n_m+1}^{n_{m+1}} \alpha_j\left(\frac{c^+(j)-c(j)}{\delta\Delta_i(m)}\right)}{\beta(m)}\right) \tag{35}$$

Now from the fact that $\sum_{j=n_m+1}^{n_{m+1}} \alpha_j/\beta_m \to 1$ as $m \to \infty$ and conclusion of Theorem 4.1 of (Bhatnagar et al., 2000), we can show that (35) exhibits the same behavior asymptotically as follows.

$$\theta_i(m+1) =$$
$$\Lambda_i\left(\theta_i(m) - \beta(m)\left(\frac{\bar{J}(\theta(m)+\delta\Delta(m)) - \bar{J}(\theta(m))}{\delta\Delta_i(m)}\right)\right). \tag{36}$$

Here $\Delta(t) = \Delta(n)$, for $t \in [t(n), t(n+1))$, where $t(n) = \sum_{k=0}^{n-1} \beta(k)$, $n \geq 1$. Now, the ODE associated with the $\theta$-update recursion (36) is as follows:

$$\dot{\theta}(t) = \bar{\Lambda}\left(-\mathbb{E}\left[\frac{\bar{J}(\theta(t)+\delta\Delta(t)) - \bar{J}(\theta(t))}{\delta\Delta_i(t)}\right]\right), \tag{37}$$

where $\mathbb{E}[\cdot]$ is with respect to the common distribution of $\Delta_i(t)$. We also consider another associated ODE

$$\dot{\theta}(t) = \bar{\Lambda}\left[-\nabla\bar{J}(\theta(t))\right], \tag{38}$$

but with the same initial condition as (37).

We recall here that the set $K$ is invariant for the ODE (38) if it is closed and any trajectory $\theta(\cdot)$ of the ODE (38) for which $\theta(0) \in K$ satisfies $\theta(t) \in K, \forall t \in \mathbb{R}$. Note that, the function $\bar{J}$ serves as a Lyapunov function for the ODE (38) since

$$\dot{\bar{J}}(\theta) = \langle\nabla\bar{J}(\theta), \dot{\theta}\rangle = \langle\nabla\bar{J}(\theta), \bar{\Lambda}(-\nabla\bar{J}(\theta))\rangle \leq 0, \ \theta \in \mathcal{C}.$$

Now, using a Taylor series expansion around the point $\theta(m)$, we get

$$\bar{J}(\theta(m)+\delta\Delta(m)) = \bar{J}(\theta(m)) + \delta\Delta(m)^T\nabla\bar{J}(\theta(m)) + O(\delta^2).$$

Hence,

$$\frac{\bar{J}(\theta(m) + \delta\Delta(m))}{\delta\Delta_i(m)} = \frac{\bar{J}(\theta(m))}{\delta\Delta_i(m)} + \nabla_i \bar{J}(\theta(m))$$

$$+ \sum_{j \neq i} \frac{\Delta_j(m)}{\Delta_i(m)} \nabla_j \bar{J}(\theta(m)) + O(\delta).$$

From our assumptions on the perturbation sequence, $\Delta(m), m \geq 0$ is zero-mean, $\pm 1$-valued Bernouli sequence. Thus, in the gradient estimation, we get the following.

$$\mathbb{E}\left[\frac{\bar{J}(\theta(m) + \delta\Delta(m)) - \bar{J}(\theta(m))}{\delta\Delta_i(m)}\right] = \nabla_i \bar{J}(\theta(m)) + O(\delta). \tag{39}$$

Now, as $\delta \to 0$, right-hand side (RHS) of (37) converges to the RHS of (38).

Therefore, it can be seen that the trajectories of the ODE (37) converge asymptotically to those of (38) uniformly over compacts for the same initial conditions. Now, $K$ is the set of asymptotically stable attractors of (38) with $\bar{J}(\cdot)$ as its associated strict Liapunov function. From the Hirsch lemma, $\|\theta_M - K\| \to 0$ a.s. as $M \to \infty$ and $\delta \to 0$. Hence, given $\gamma > 0, \exists \delta_0 > 0$, s.t. $\forall \delta \in (0, \delta_0]$, $\theta_M \to \theta^* \in K^\gamma$ a.s. as $M \to \infty$.

$\square$

### A.1.2 Asymptotic Analysis of Algorithm 2

**Proof of Lemma 2:**

*Proof.* Recall that

$$\mathcal{L}_{\pi^\theta, \eta} = \lim_{n \to \infty} \frac{1}{n} \mathbb{E}[\sum_{k=0}^{n-1} h_{k+1} \mid \pi^\theta] = \sum_{s \in \mathcal{S}} d_{\pi^\theta}(s) \sum_{a \in \mathcal{A}} \pi^\theta(s, a) h(s, a)$$

$$= \sum_{s \in S} d_{\pi^\theta}(s) \sum_{a \in \mathcal{A}(s)} \pi^\theta(s, a)[r(s, a) - \sum_{q=1}^N \eta_q (g_q(s, a) - \nu_q)],$$

where $d_{\pi^\theta} = (d_{\pi^\theta}(s), s \in \mathcal{S})$ is the stationary probability distribution of the ergodic Markov process $\{\mathcal{X}_n, n \geq 0\}$. As the expected value of single-stage reward $r(s, a)$ and costs $g_q, q = 1, \ldots, N$ are uniformly bounded and from the definition of $\pi^\theta(s, a)$, see (6), we can check that $\pi^\theta(s, a)$ is continuously differentiable with respect to $\theta$. Further, as in proof of Lemma 1, $d_{\pi^\theta}$ is continuously differentiable. Thus, $\mathcal{L}_{\pi^\theta}$ is continuously differentiable in $\theta \in \mathcal{C}$. $\square$

We define set $K''$ as follows:

$$K'' = \{\theta \in \mathcal{C} \mid \bar{\Lambda}\left[-\nabla \bar{\mathcal{L}}(\theta)\right] = 0\}, \tag{40}$$

where $\bar{\Lambda}$ is as in (15). Further, given $\gamma > 0$, let $K_\gamma$ denote the $\gamma$-neighborhood of the set $K''$, i.e.,

$$K_\gamma = \{\theta \in \mathcal{C} \mid \|\theta - \theta_0\| < \gamma, \ \theta_0 \in K''\}.$$

**Proof of Theorem 2:**

*Proof.* Note that the (17) can be rewritten as:

$$\theta_i(m+1) = \Lambda_i \left(\theta_i(m) - \beta(m)\frac{\sum_{j=n_m+1}^{n_{m+1}} \alpha_j \left(\frac{\bar{h}^+(j) - \bar{h}(j)}{\delta\Delta_i(m)}\right)}{\beta(m)}\right) \tag{41}$$

Now from the fact that $\sum_{j=n_m+1}^{n_{m+1}} \alpha_j / \beta(m) \to 1$ as $m \to \infty$ and conclusion of Theorem 4.1 of (Bhatnagar et al., 2000), we can show that (41) exhibit the same behavior asymptotically as follows.

$$\theta_i(m+1) = \Lambda_i \left( \theta_i(m) - \beta(m) \left( \frac{\bar{\mathcal{L}}(\theta(m) + \delta\Delta(m), \eta) - \bar{\mathcal{L}}(\theta(m), \eta)}{\delta\Delta_i(m)} \right) \right). \tag{42}$$

Here $\Delta(t) = \Delta(n)$, for $t \in [t(n), t(n+1))$, where $t(n) = \sum_{k=0}^{n-1} \beta(k)$, $n \geq 1$. Now, the ODE associated with the $\theta$-update recursion, i.e., (42) is as follows:

$$\dot{\theta}(t) = \bar{\Lambda} \left( -\mathbb{E} \left[ \frac{\bar{\mathcal{L}}(\theta(t) + \delta\Delta(t), \eta) - \bar{\mathcal{L}}(\theta(t), \eta)}{\delta\Delta_i(t)} \right] \right), \tag{43}$$

where $\mathbb{E}[\cdot]$ is with respect to the common distribution of $\Delta_i(t)$. We also consider another associated ODE

$$\dot{\theta}(t) = \Lambda \left[ -\nabla\bar{\mathcal{L}}(\theta(t), \eta) \right], \tag{44}$$

having the same initial condition as (43).

Now denote by $K^\eta$ the largest invariant set contained within the set $K''$. We recall here that the set $K^\eta$ is invariant for the ODE (44) if it is closed and any trajectory $\theta(\cdot)$ of the ODE (44) for which $\theta(0) \in K^\eta$ satisfies $\theta(t) \in K^\eta$, $\forall t \in \mathbb{R}$. Note that, the function $\bar{\mathcal{L}}$ serves as a Lyapunov function for the ODE (44) since

$$\dot{\bar{\mathcal{L}}}(\theta, \eta) = \langle \nabla\bar{\mathcal{L}}(\theta, \eta), \dot{\theta} \rangle = -\|\nabla\bar{\mathcal{L}}(\theta, \eta)\|^2$$

Thus,

$$\dot{\bar{\mathcal{L}}}(\theta, \eta) \quad < 0 \quad \forall \theta \notin K''$$
$$= 0 \quad \text{otherwise.}$$

Given $\gamma > 0$, let $K_\gamma^\eta$ be the set of points within a distance $\gamma$ from the points in the set $K^\eta$, i.e.,

$$K_\gamma^\eta = \{\theta \in \mathcal{C} \mid \|\theta - \theta_0\| < \gamma, \theta_0 \in K^\eta\}. \tag{45}$$

Now, using the Taylor series expansion around the point $\theta(m)$ we get

$$\bar{\mathcal{L}}(\theta(m) + \delta\Delta(m), \eta) = \bar{\mathcal{L}}(\theta(m), \eta) + \delta\Delta(m)^T \nabla\bar{\mathcal{L}}(\theta(m), \eta) + O(\delta^2).$$

Hence,

$$\frac{\bar{\mathcal{L}}(\theta(m) + \delta\Delta(m), \eta)}{\delta\Delta_i(m)} = \frac{\bar{\mathcal{L}}(\theta(m), \eta)}{\delta\Delta_i(m)} + \nabla_i\bar{\mathcal{L}}(\theta(m), \eta) + \sum_{j \neq i} \frac{\Delta_j(m)}{\Delta_i(m)} \nabla_j\bar{\mathcal{L}}(\theta(m), \eta) + O(\delta).$$

From our assumptions on the perturbation sequence, $\Delta(m), m \geq 0$ is zero-mean, $\pm 1$-valued Bernoulli sequence. Thus, in the gradient estimation, we get the following.

$$\mathbb{E} \left[ \frac{\bar{\mathcal{L}}(\theta(m) + \delta\Delta(m), \eta) - \bar{\mathcal{L}}(\theta(m), \eta)}{\delta\Delta_i(m)} \right] = \nabla_i\bar{\mathcal{L}}(\theta(m), \eta) + O(\delta). \tag{46}$$

Now, as $\delta \to 0$, right-hand side (RHS) of (43) converges to the RHS of (44). Therefore, it can be seen that the trajectories of the ODE (43) converge asymptotically to those of (44) uniformly over compacts for the same initial conditions. Now, $K^\eta$ is the set of asymptotically stable attractors of (44) with $\bar{\mathcal{L}}(\cdot)$ as its associated strict Liapunov function. From the Hirsch lemma, $\|\theta_M - K^\eta\| \to 0$ a.s. as $M \to \infty$ and $\delta \to 0$. Hence, given $\gamma > 0, \exists \delta_0 > 0$, s.t. $\forall \delta \in (0, \delta_0], \theta_M \to \theta^* \in K_\gamma^\eta$ a.s. as $M \to \infty$. $\qquad \square$

Now, define $E := \{\eta = (\eta_1, \ldots, \eta_N)^\top \mid \eta_q \in [0, M_\eta], \hat{\bar{\Lambda}}(\mathcal{G}_q(\theta^\eta) - \nu_q) = 0, \forall q = 1, \ldots, N, \theta^\eta \in K^\eta\}.$

**Proposition 1.** *Under the Assumptions 1,2 and $\theta(n) \equiv \theta$, $\lim_{n\to\infty} \eta(n) = \eta^*$ with probability one, for some $\eta^* = [\eta_1^*, \cdots, \eta_N^*]^\top \in E$.*

*Proof.* Under the Assumptions 1,2 and already proven $\theta(n) \equiv \theta, \forall n$, we get $\lim_{n\to\infty} \mathcal{G}_q(n) = \mathcal{G}_q(\theta), q = 1, \ldots, N$ with probability one as in Proposition 4.2 of (Bhatnagar & Lakshmanan, 2012). Further, the proof sketch is similar to the proof of Theorem 4.3 of (Bhatnagar & Lakshmanan, 2012). $\square$

## A.2 Finite Time Analysis

### A.2.1 Finite Time Analysis of Algorithm 1

**Proof of Lemma 3:**

$$\sum_{j=n_m+1}^{n_{m+1}} \alpha_j = \sum_{j=n_m+1}^{n_{m+1}} \frac{1}{(1+j)^{\sigma'''}} \leq \sum_{j=n_m+1}^{n_{m+1}} \frac{1}{j^{\sigma'''}} = \frac{1}{\{n_m+1\}^{\sigma'''}} + \frac{1}{\{n_m+2\}^{\sigma'''}} + \cdots + \frac{1}{\{n_{m+1}\}^{\sigma'''}}$$

Therefore,

$$\frac{\sum_{j=n_m+1}^{n_{m+1}} \alpha_j}{\beta_{m_n}} \leq \frac{\frac{1}{\{n_m+1\}^{\sigma'''}} + \frac{1}{\{n_m+2\}^{\sigma'''}} + \cdots + \frac{1}{\{n_{m+1}\}^{\sigma'''}}}{\frac{1}{\{n_m+1\}^{\sigma''}}}$$

$$= \{n_m+1\}^{\sigma''} \left[ \frac{1}{\{n_m+1\}^{\sigma'''}} + \frac{1}{\{n_m+2\}^{\sigma'''}} + \cdots + \frac{1}{\{n_{m+1}\}^{\sigma'''}} \right]$$

$$\leq \{n_m+1\}^{\sigma''} \left[ \frac{1}{\{n_m+1\}^{\sigma'''}} + \frac{1}{\{n_m+1\}^{\sigma'''}} + \cdots + \frac{1}{\{n_m+1\}^{\sigma'''}} \right]$$

$$\leq \{n_m+1\}^{\sigma''} \frac{c''\{n_m\}^{\sigma'}}{\{n_m+1\}^{\sigma'''}} \text{ as per the assumption}$$

$$\leq \frac{c''\{n_m+1\}^{\sigma''+\sigma'}}{\{n_m+1\}^{\sigma'''}} \text{ as } \{n_m\}^{\sigma'} \leq \{n_m+1\}^{\sigma'}$$

$$= \frac{c''}{\{n_m+1\}^{\sigma'''-\sigma''-\sigma'}}$$

$$\text{i.e., } \frac{\sum_{j=n_m+1}^{n_{m+1}} \alpha_j}{\beta_{m_n}} \leq \frac{c''}{\{n_m+1\}^{\sigma'''-\sigma''-\sigma'}} \tag{47}$$

From $0 < \sigma' + \sigma'' < \sigma'''$, we get $\sigma''' - \sigma'' - \sigma' > 0$. Now if $n_m = 0$ then $\frac{\sum_{j=n_m+1}^{n_{m+1}} \alpha_j}{\beta_{m_n}} \leq c''$ and if $n_m$ is large number then $\frac{\sum_{j=n_m+1}^{n_{m+1}} \alpha_j}{\beta_{m_n}}$ tends to 0. Hence, the above claim is satisfied and $\max_m \frac{\sum_{j=n_m+1}^{n_{m+1}} \alpha_j}{\beta(m)}$ is bounded. $\square$

**Proof of Lemma 4:**

$$\mathbb{E}_m \left[ \mathcal{N}(\theta_i(m)] = \mathbb{E}_m \left[ \frac{1}{\delta\Delta_i(m)} \left[ \frac{\sum_{j=n_m+1}^{n_{m+1}} \alpha_j \left( c^+(j) - c(j) \right)}{\beta(m)} - \left( \bar{J}(\theta(m) + \delta\Delta(m)) - \bar{J}(\theta(m)) \right) \right] \right]$$

$$\leq \frac{1}{\delta} \mathbb{E}_m \left[ \frac{\sum_{j=n_m+1}^{n_{m+1}} \alpha_j \left( c^+(j) - c(j) \right)}{\beta(m)} - \left( \bar{J}(\theta(m) + \delta\Delta(m)) - \bar{J}(\theta(m)) \right) \right]$$

Now,

$$\mathbb{E}_m \left[ \frac{\sum_{j=n_m+1}^{n_{m+1}} \alpha_j \left( c^+(j) - c(j) \right)}{\beta(m)} - \left( \bar{J}(\theta(m) + \delta\Delta(m)) - \bar{J}(\theta(m)) \right) \right]$$

$$\leq \mathbb{E}_m \left[ \{\max_m \frac{\sum_{j=n_m+1}^{n_{m+1}} \alpha_j}{\beta(m)}\} \frac{\sum_{j=n_m+1}^{n_{m+1}} \alpha_j \left( c^+(j) - c(j) \right)}{\sum_{j=n_m+1}^{n_{m+1}} \alpha_j} - \left( \bar{J}(\theta(m) + \delta\Delta(m)) - \bar{J}(\theta(m)) \right) \right]$$

$$\leq \mathbb{E}_m \left[ \frac{c''}{\{1+n_m\}^{\sigma'''-\sigma''-\sigma'}} \frac{1}{\sum_{j=n_m+1}^{n_{m+1}} \alpha_j} \sum_{j=n_m+1}^{n_{m+1}} \alpha_j [(c^+(j) - \bar{J}(\theta(m) + \delta\Delta(m))) - (c(j) - \bar{J}(\theta(m)))] \right]$$

(from Lemma 3, see(47))

$$\leq \frac{c''}{\{1+n_m\}^{\sigma'''-\sigma''-\sigma'}} \frac{1}{\sum_{j=n_m+1}^{n_{m+1}} \alpha_j} \sum_{j=n_m+1}^{n_{m+1}} 4\alpha_j \vartheta \rho^{m-1} B_8 (\text{using Assumption } 3, 0 < \rho < 1, B_8 > 0)$$

$$\leq \frac{c''}{\{1+n_m\}^{\sigma'''-\sigma''-\sigma'}} 4B_8 \frac{1}{\sum_{j=n_m+1}^{n_{m+1}} \alpha_j} \sum_{j=n_m+1}^{n_{m+1}} \alpha_j \beta_{n_m} = \frac{B_1}{\{1+n_m\}^{\sigma'''-\sigma''-\sigma'}} \beta_{n_m}$$

$$= \frac{B_1}{\{1+n_m\}^{\sigma'''-\sigma''-\sigma'}} \frac{1}{\{1+n_m\}^{\sigma''}} = \frac{B_1}{\{1+n_m\}^{\sigma'''-\sigma'}} \leq \frac{B_1}{\{1+n_m\}^{\sigma''}} \leq B_1 \beta_{m_n}$$

In the above, $B_1 = 4B_8 c''$ and from Assumption 3, $m \geq \iota_m$ where $\iota_m$ is mixing time, $\vartheta \rho^{m-1} \leq \beta(m)$.

Thus, $\mathbb{E}_m[\|\mathcal{N}(\theta(m)\|] \leq \frac{B_1 \beta_m}{\delta}$.

In the similar way we can show that $\mathbb{E}_m[\|\mathcal{N}(\theta(m)\|^2] \leq \frac{B_4 \beta_m^2}{\delta^2}$.  □

**Proof of Lemma 5:**
Let, assume that gradient of $\bar{J}(\cdot)$, i.e., $\nabla \bar{J}(\cdot)$ is estimated by $\hat{\nabla} \bar{J}(\cdot)$, and from (21), (22) we know that

$$\hat{\nabla}_i \bar{J}(\theta(m)) = \frac{\bar{J}(\theta(m) + \delta\Delta(m)) - \bar{J}(\theta(m))}{\delta\Delta_i(m)}. \tag{48}$$

Thus,

$$\mathbb{E}\left[ \frac{\bar{J}(\theta(m) + \delta\Delta(m)) - \bar{J}(\theta(m))}{\delta\Delta_i(m)} \mid \theta(m) \right] = \nabla_i \bar{J}(\theta(m)) + c_1 \delta, \tag{49}$$

for some constant term $c_1 > 0$. Now,

$$\|\nabla \bar{J}(\theta)\|_1 = \sum_{i=1}^{d} |\nabla_i \bar{J}(\theta)| = \sum_{i=1}^{d} \left| \mathbb{E}\left[ \frac{\bar{J}(\theta + \delta\Delta) - \bar{J}(\theta)}{\delta\Delta_i} \mid \theta \right] - c_1 \delta \right|$$

$$\leq \sum_{i=1}^{d} \left| \mathbb{E}\left[ \frac{\bar{J}(\theta + \delta\Delta) - \bar{J}(\theta)}{\delta\Delta_i} \mid \theta \right] \right| + |c_1 \delta| \leq B.$$

The last inequality holds as single-stage rewards are bounded and hence, $\bar{J}(\cdot)$ is bounded.

□

**Proof of Lemma 6:**
From (49), as in (A. & Bhatnagar, 2024), it is easy to see that the proof holds.  □

**Definition 2.** *L-smooth function:* *A function $f : \mathcal{C} \subset \mathbb{R}^d \to \mathbb{R}$ be L-smooth if for some $L > 0$, $\forall$ $\theta', \theta'' \in \mathcal{C}$, $f(\cdot)$ satisfies $\|\nabla f(\theta') - \nabla f(\theta'')\| \leq L \|\theta' - \theta''\|$.*

**Proof of Theorem 3:**

Since $\bar{J}$ is $L$-smooth, (see Definition 2), as in (A. & Bhatnagar, 2024; Papini et al., 2018), we have

$$
\bar{J}(\theta_{k+1}) \leq \bar{J}(\theta_k) + \langle \nabla \bar{J}(\theta_k), \theta_{k+1} - \theta_k \rangle + \frac{L}{2} \|\theta_{k+1} - \theta_k\|^2
$$

$$
\leq \bar{J}(\theta_k) - \beta_k \left\langle \nabla \bar{J}(\theta_k), \widehat{\nabla} \bar{J}(\theta_k) + \mathcal{N}(\theta_k) \right\rangle + \frac{L}{2} \beta_k^2 \left\| \widehat{\nabla} \bar{J}(\theta_k) + \mathcal{N}(\theta_k) \right\|^2
$$

$$
\leq \bar{J}(\theta_k) - \beta_k \left\langle \nabla \bar{J}(\theta_k), \widehat{\nabla} \bar{J}(\theta_k) \right\rangle - \beta_k \left\langle \nabla \bar{J}(\theta_k), \mathcal{N}(\theta_k) \right\rangle + \frac{L}{2} \beta_k^2 \left[ \left\| \widehat{\nabla} \bar{J}(\theta_k) \right\|^2 + \|\mathcal{N}(\theta_k)\|^2 \right] \quad (50)
$$

Taking expectations with respect to the sigma field $\mathcal{F}_k$ on both sides of (50), we obtain

$$
\mathbb{E}_k \left[ \bar{J}(\theta_{k+1}) \right] \leq \mathbb{E}_k \left[ \bar{J}(\theta_k) \right] - \beta_k \left\langle \nabla \bar{J}(\theta_k), \nabla \bar{J}(\theta_k) + c_1 \delta \mathbf{1}_{d \times 1} \right\rangle - \beta_k B \mathbb{E}_k \|\mathcal{N}(\theta_k)\|
$$

$$
+ \frac{L}{2} \beta_k^2 \left[ \left\| \mathbb{E}_k \left[ \widehat{\nabla} \bar{J}(\theta_k) \right] \right\|^2 + \frac{c_2}{\delta^2} \right] + \frac{L}{2} \beta_k^2 \|\mathcal{N}(\theta_k)\|^2
$$

$$
\leq \bar{J}(\theta_k) - \beta_k \left\| \nabla \bar{J}(\theta_k) \right\|^2 + c_1 \delta \beta_k \mathbb{E}_k \left\| \nabla \bar{J}(\theta_k) \right\|_1 - B \beta_k \frac{B_1 \beta_k}{\delta}
$$

$$
+ \frac{L}{2} \beta_k^2 \left[ \left\| \nabla \bar{J}(\theta_k) + c_1 \delta \mathbf{1}_{d \times 1} \right\|^2 + \frac{c_2}{\delta^2} \right] + \frac{L}{2} \beta_k^2 \frac{B_4 \beta_k^2}{\delta^2} \quad (51)
$$

$$
\leq \bar{J}(\theta_k) - \beta_k \left\| \nabla \bar{J}(\theta_k) \right\|^2 + c_1 \delta \beta_k \mathbb{E}_k \left\| \nabla \bar{J}(\theta_k) \right\|_1 - \frac{B B_1 \beta_k^2}{\delta}
$$

$$
+ \frac{L}{2} \beta_k^2 \left[ \left\| \nabla \bar{J}(\theta_k) \right\|^2 + 2 c_1 \delta \mathbb{E}_k \left\| \nabla \bar{J}(\theta_k) \right\|_1 + d c_1^2 \delta^2 + \frac{c_2}{\delta^2} \right] + \frac{L}{2 \delta^2} B_4 \beta_k^4
$$

$$
\leq \bar{J}(\theta_k) - \left( \beta_k - \frac{L}{2} \beta_k^2 \right) \left\| \nabla \bar{J}(\theta_k) \right\|^2 + c_1 \delta B \left( \beta_k + L \beta_k^2 \right) - \frac{B B_1 \beta_k^2}{\delta} + \frac{L}{2} \beta_k^2 \left[ d c_1^2 \delta^2 + \frac{c_2}{\delta^2} \right] + \frac{L}{2 \delta^2} B_4 \beta_k^4, \quad (52)
$$

The 1st inequality follows from (23), (24) in Lemma 6, and from Lemma 4 . In the above, $-\|y\|_1 \leq \sum_{i=1}^{d} y_i$ for any $d$-vector $y$, is used to get the inequality in (51). The last inequality follows from the fact that $\left\| \nabla \bar{J}(\theta_k) \right\|_1 \leq B$ by Lemma 5. Now, rearranging the terms,

$$
\left\| \nabla \bar{J}(\theta_k) \right\|^2 \leq \frac{2}{\beta_k(2 - L\beta_k)} \left[ \bar{J}(\theta_k) - \mathbb{E}_k \bar{J}(\theta_{k+1}) + c_1 \delta \left( \beta_k + L \beta_k^2 \right) B \right]
$$

$$
+ \frac{L \beta_k^2}{\beta_k(2 - L\beta_k)} \left[ d c_1^2 \delta^2 + \frac{c_2}{\delta^2} \right] + \frac{2 \beta_k^2}{\beta_k(2 - L\beta_k)} \left[ \frac{L B_4}{2 \delta^2} \beta_k^2 - \frac{B B_1}{\delta} \right]
$$

Now, as in (Wu et al., 2020), we sum up the inequality above for $k = \iota_m$ to $m$, take expectations, divide by $(1 + m - \iota_m)$ both sides and assume $m > 2\iota_m - 1$. We now obtain the following.

$$
\frac{1}{1 + m - \iota_m} \sum_{k=\iota_m}^{m} \mathbb{E}_k \left\| \nabla \bar{J}(\theta_k) \right\|^2
$$

$$
\leq \frac{2}{1 + m - \iota_m} \sum_{k=\iota_m}^{m} \frac{\left( \mathbb{E}_k \bar{J}(\theta_k) - \mathbb{E}_k \bar{J}(\theta_{k+1}) \right)}{\beta_k(2 - L\beta_k)} + \frac{2}{1 + m - \iota_m} \sum_{k=\iota_m}^{m} c_1 \delta B \left( \frac{1 + L\beta_k}{2 - L\beta_k} \right)
$$

$$
+ \frac{L}{1 + m - \iota_m} \sum_{k=\iota_m}^{m} \frac{\beta_k}{(2 - L\beta_k)} \left[ d c_1^2 \delta^2 + \frac{c_2}{\delta^2} \right] + \frac{2}{1 + m - \iota_m} \sum_{k=\iota_m}^{m} \frac{\beta_k}{(2 - L\beta_k)} \left[ \frac{L B_4}{2 \delta^2} \beta_k^2 - \frac{B B_1}{\delta} \right] \quad (53)
$$

Now, we denote 1st, 2nd, 3rd and 4th terms of right-hand-side of (53) as $I_1$, $I_2$, $I_3$, and $I_4$ respectively.

In $I_1$,

$$\sum_{k=\iota_m}^{m} \frac{1}{\beta_k} * \frac{\left(\mathbb{E}_k \bar{J}(\theta_k) - \mathbb{E}_k \bar{J}(\theta_{k+1})\right)}{(2 - L\beta_k)} \le \sum_{k=\iota_m}^{m} \frac{1}{\beta_k} * \left(\mathbb{E}_k \bar{J}(\theta_k) - \mathbb{E}_k \bar{J}(\theta_{k+1})\right)$$

$$= \sum_{k=\iota_m}^{m} \left(\frac{1}{\beta_k} - \frac{1}{\beta_{k-1}}\right) \mathbb{E}_k\left[\bar{J}(\theta_k)\right] + \frac{1}{\beta_{\iota_m-1}} \mathbb{E}_k\left[\bar{J}(\theta_{\iota_m})\right] - \frac{1}{\beta_m} \mathbb{E}_k\left[\bar{J}(\theta_{m+1})\right]$$

$$\le \sum_{k=\iota_m}^{m} \left(\frac{1}{\beta_k} - \frac{1}{\beta_{k-1}}\right) B_r + \frac{1}{\beta_{\iota_m-1}} B_r - \frac{1}{\beta_m} B_r \le B_r \left[\sum_{k=\iota_m}^{m} \left(\frac{1}{\beta_k} - \frac{1}{\beta_{k-1}}\right) + \frac{1}{\beta_{\iota_m-1}}\right] = 2B_r\beta_m^{-1},$$

In the above, the 1st inequality is due to $\beta_k \le \frac{1}{L}$. The 2nd inequality holds due to $|\mathbb{E}_k[\bar{J}(\theta_k)]| \le B_r$ as single stage rewards $r$ are bounded.

From $I_2$,

$$\frac{2}{1+m-\iota_m} \sum_{k=\iota_m}^{m} c_1 \delta B \left(\frac{1+L\beta_k}{2-L\beta_k}\right) \le \frac{2}{1+m-\iota_m} \sum_{k=\iota_m}^{m} c_1 \delta B (1+L\beta_k)$$

$$\le \frac{2}{1+m-\iota_m} \sum_{k=\iota_m}^{m} 2c_1 \delta B \beta_k \le B_5 \beta_m = \mathcal{O}(\frac{1}{m^{1/2}})$$

In the above the 1st and 2nd inequality is due to $\beta_k \le \frac{1}{L}$ and $B_5 > 0$ some constant term.

From $I_3$ we get,

$$\frac{L}{1+m-\iota_m} \sum_{k=\iota_m}^{m} \frac{\beta_k}{(2-L\beta_k)} \left[dc_1^2\delta^2 + \frac{c_2}{\delta^2}\right] \le \frac{L}{1+m-\iota_m} \sum_{k=\iota_m}^{m} \beta_k \left[dc_1^2\delta^2 + \frac{c_2}{\delta^2}\right]$$

$$\le B_6 \beta_m = \mathcal{O}(\frac{1}{m^{1/2}})$$

In the above, first inequality is due to $\beta_k \le \frac{1}{L}$, and a constant $B_6 > 0$.

Further, From $I_4$

$$\frac{2}{1+m-\iota_m} \sum_{k=\iota_m}^{m} \frac{\beta_k}{(2-L\beta_k)} \left[\frac{LB_4}{2\delta^2}\beta_k^2 - \frac{BB_1}{\delta}\right]$$

$$\le \frac{2}{1+m-\iota_m} \sum_{k=\iota_m}^{m} \beta_k \left[\frac{LB_4}{2\delta^2}\beta_k^2 - \frac{BB_1}{\delta}\right] \le B_7 \beta_m^3 = \mathcal{O}(\frac{1}{m^{3/2}})$$

In the above, first inequality is due to $\beta_k \le \frac{1}{L}$, and a constant $B_7 > 0$.

Now from (53)

$$\min_{0 \le k \le m} \mathbb{E}\left\|\nabla\bar{J}(\theta_k)\right\|^2 = \frac{1}{1+m-\iota_m} \sum_{k=\iota_m}^{m} \mathbb{E}_k\left\|\nabla\bar{J}(\theta_k)\right\|^2$$

$$\le \frac{4B_r\beta_m^{-1}}{1+m-\iota_m} + \mathcal{O}(\frac{1}{m^{1/2}}) + \mathcal{O}(\frac{1}{m^{3/2}})$$

$$= \mathcal{O}(\frac{1}{m^{1/2}}) + \mathcal{O}(\frac{1}{m^{1/2}}) = \mathcal{O}(\epsilon^{-2})$$

$\square$

### A.2.2 Finite Time Analysis of Algorithm 2

**Lemma 7.** *The faster and slower than faster step sizes* $\beta_n = \frac{1}{\{1+n\}^{\sigma_4}}$, $\alpha_n = \frac{1}{\{1+n\}^{\sigma_5}}$, *where* $0 < \sigma_4 < \sigma_5$, *follow*

$$0 \le \frac{\sum_{j=n_m+1}^{n_{m+1}} \alpha_j}{\beta_{m_n}} \le c''', \tag{54}$$

*assuming* $0 \le n_{m+1} - n_m \le c'''\{n_m\}^{\sigma_3}$, $c''' > 0$ *where* $0 < \sigma_4 + \sigma_3 < \sigma_5$. *Thus,* $\frac{\sum_{j=n_m+1}^{n_{m+1}} \alpha_j}{\beta_{m_n}}$ *is bounded.*

*Proof.* The proof follows by replacing $\sigma'''$, $\sigma''$, $\sigma'$ with $\sigma_5$, $\sigma_4$ and $\sigma_3$, respectively, in the proof of Lemma 3. $\qquad\square$

**Lemma 8.** $\mathbb{E}_m[\|\mathcal{N}_1(\theta(m), \eta(m)\|] \le \frac{B_{10}\beta_m}{\delta}$ *and* $\mathbb{E}_m[\|\mathcal{N}_1(\theta(m), \eta(m)\|^2] \le \frac{B_{14}\beta_m^2}{\delta^2}$ *for some constant* $B_{10}, B_{14} > 0$.

*Proof.*

$$\mathbb{E}_m\left[\mathcal{N}_1(\theta_i(m), \eta(m)\right]$$
$$= \mathbb{E}_m\left[\frac{1}{\delta \Delta_i(m)}\left[\frac{\sum_{j=n_m+1}^{n_{m+1}} \alpha_j\left(\bar{h}^+(j) - \bar{h}(j)\right)}{\beta(m)} - \left(\bar{\mathcal{L}}(\theta(m) + \delta\Delta(m), \eta(m)) - \bar{\mathcal{L}}(\theta(m), \eta(m))\right)\right]\right]$$
$$\le \frac{1}{\delta}\mathbb{E}_m\left[\frac{\sum_{j=n_m+1}^{n_{m+1}} \alpha_j\left(\bar{h}^+(j) - \bar{h}(j)\right)}{\beta(m)} - \left(\bar{\mathcal{L}}(\theta(m) + \delta\Delta(m), \eta(m)) - \bar{\mathcal{L}}(\theta(m), \eta(m))\right)\right]$$

Now,

$$\mathbb{E}_m\left[\frac{\sum_{j=n_m+1}^{n_{m+1}} \alpha_j\left(\bar{h}^+(j) - \bar{h}(j)\right)}{\beta(m)} - \left(\bar{\mathcal{L}}(\theta(m) + \delta\Delta(m), \eta(m)) - \bar{\mathcal{L}}(\theta(m), \eta(m))\right)\right]$$

$$\le \mathbb{E}_m[\{\max_m \frac{\sum_{j=n_m+1}^{n_{m+1}} \alpha_j}{\beta(m)}\}\frac{\sum_{j=n_m+1}^{n_{m+1}} \alpha_j\left(\bar{h}^+(j) - \bar{h}(j)\right)}{\sum_{j=n_m+1}^{n_{m+1}} \alpha_j} - (\bar{\mathcal{L}}(\theta(m) + \delta\Delta(m), \eta(m)) -$$

$$\bar{\mathcal{L}}(\theta(m), \eta(m)))] \le \mathbb{E}_m[\frac{c'''}{\{1 + n_m\}^{\sigma_5 - \sigma_4 - \sigma_3}}\frac{1}{\sum_{j=n_m+1}^{n_{m+1}} \alpha_j}\sum_{j=n_m+1}^{n_{m+1}} \alpha_j$$

$$[(\bar{h}^+(j) - \bar{\mathcal{L}}(\theta(m) + \delta\Delta(m), \eta(m))) - (\bar{h}(j) - \bar{\mathcal{L}}(\theta(m), \eta(m)))]] \text{ (from Lemma 7, as in (47))}$$

$$\le \frac{c'''}{\{1 + n_m\}^{\sigma_5 - \sigma_4 - \sigma_3}}\frac{1}{\sum_{j=n_m+1}^{n_{m+1}} \alpha_j}\sum_{j=n_m+1}^{n_{m+1}} 4\alpha_j\vartheta\rho^{m-1}B_{11} \text{ (using Assumption 3, } 0 < \rho < 1, B_{11} > 0 \text{ constant)}$$

$$\le \frac{c'''}{\{1 + n_m\}^{\sigma_5 - \sigma_4 - \sigma_3}}4B_{11}\frac{1}{\sum_{j=n_m+1}^{n_{m+1}} \alpha_j}\sum_{j=n_m+1}^{n_{m+1}} \alpha_j\beta_{n_m} = \frac{B_{10}}{\{1 + n_m\}^{\sigma_5 - \sigma_4 - \sigma_3}}\beta_{n_m}$$

$$= \frac{B_{10}}{\{1 + n_m\}^{\sigma_5 - \sigma_4 - \sigma_3}}\frac{1}{\{1 + n_m\}^{\sigma_4}} = \frac{B_{10}}{\{1 + n_m\}^{\sigma_5 - \sigma_3}} \le \frac{B_{10}}{\{1 + n_m\}^{\sigma_5 - \sigma_3}} \le \frac{B_{10}}{\{1 + n_m\}^{\sigma_4}} \le B_{10}\beta_{m_n}$$

In the above, $B_{10} = 4B_{11}c'''$ and from Assumption 3, $m \ge \iota_m$ where $\iota_m$ is mixing time, $\vartheta\rho^{m-1} \le \beta(m)$. Thus, $\mathbb{E}_m[\|\mathcal{N}_1(\theta(m), \eta(m)\|] \le \frac{B_{10}\beta_m}{\delta}$.

In the similar way we can show that $\mathbb{E}_m[\|\mathcal{N}_1(\theta(m), \eta(m)\|^2] \le \frac{B_{14}\beta_m^2}{\delta^2}$. $\qquad\square$

**Lemma 9.** *There exists a constant* $B' > 0$ *such that* $\|\nabla\bar{\mathcal{L}}(\theta), \eta(m)\|_1 \le B', \forall \theta \in \mathbb{R}^d$.

*Proof.* Let, assume that gradient of $\bar{\mathcal{L}}(\cdot)$, i.e., $\nabla\bar{\mathcal{L}}(\cdot)$ is estimated by $\hat{\nabla}\bar{\mathcal{L}}(\cdot)$, and from (26), (27) we know that $\hat{\nabla}_i\bar{\mathcal{L}}(\theta(m),\eta(m)) = \frac{\bar{\mathcal{L}}(\theta(m)+\delta\Delta(m),\eta(m))-\bar{\mathcal{L}}(\theta(m),\eta(m))}{\delta\Delta_i(m)}$. Thus,

$$\mathbb{E}\left[\frac{\bar{\mathcal{L}}(\theta(m)+\delta\Delta(m),\eta(m))-\bar{\mathcal{L}}(\theta(m),\eta(m))}{\delta\Delta_i(m)} \mid \theta(m)\right] = \nabla_i\bar{\mathcal{L}}(\theta(m),\eta(m)) + c_3\delta, \tag{55}$$

for some constant term $c_3 > 0$. Now,

$$\|\nabla\bar{\mathcal{L}}(\theta),\eta(m)\|_1 = \sum_{i=1}^d \mid \nabla_i\bar{\mathcal{L}}(\theta,\eta(m)) \mid = \sum_{i=1}^d \mid \mathbb{E}\left[\frac{\bar{\mathcal{L}}(\theta+\delta\Delta,\eta(m))-\bar{\mathcal{L}}(\theta,\eta(m))}{\delta\Delta_i} \mid \theta\right] - c_3\delta \mid$$

$$\leq \sum_{i=1}^d \mid \mathbb{E}\left[\frac{\bar{\mathcal{L}}(\theta+\delta\Delta,\eta(m))-\bar{\mathcal{L}}(\theta,\eta(m))}{\delta\Delta_i} \mid \theta\right] \mid + \mid c_3\delta \mid \leq B'.$$

The last inequality holds as the expected value of single-stage reward $r$ and costs $g_q, q = 1,\ldots,N$ in Algorithm 2 are uniformly bounded and hence, $\bar{\mathcal{L}}(\cdot)$ is bounded.

$\square$

**Lemma 10.** *The gradient estimate $\hat{\nabla}\bar{\mathcal{L}}(\theta_k,\eta(k))$ satisfies the following inequalities for all $k \geq 1$ :*

$$\left\|\mathbb{E}_k\left[\hat{\nabla}\bar{\mathcal{L}}(\theta_k,\eta(k))\right] - \nabla\bar{\mathcal{L}}(\theta_k,\eta(k))\right\|_\infty \leq c_3\delta \tag{56}$$

*and*

$$\mathbb{E}_k\left[\left\|\hat{\nabla}\bar{\mathcal{L}}(\theta_k,\eta(k))\right\|^2\right] \leq \left\|\mathbb{E}_k\left[\hat{\nabla}\bar{\mathcal{L}}(\theta_k,\eta(k))\right]\right\|^2 + \frac{c_4}{\delta^2} \tag{57}$$

In the above, $\mathbb{E}_k$ is shorthand for $\mathbb{E}(\cdot \mid \mathcal{F}_k)$, with sigma-field $\mathcal{F}_k$ and $c_3, c_4$ are some positive constants.

*Proof.* From (55), as in (A. & Bhatnagar, 2024), it is easy to see that the proof holds. $\square$

**Proof of Theorem 4:**
Since $\bar{\mathcal{L}}$ is $L$-smooth, (see Definition 2), as in (A. & Bhatnagar, 2024; Papini et al., 2018), we have

$$\bar{\mathcal{L}}(\theta_{k+1},\eta(k)) \leq \bar{\mathcal{L}}(\theta_k,\eta(k)) + \left\langle\nabla\bar{\mathcal{L}}(\theta_k,\eta(k)),\theta_{k+1}-\theta_k\right\rangle + \frac{L}{2}\|\theta_{k+1}-\theta_k\|^2$$

$$\leq \bar{\mathcal{L}}(\theta_k,\eta(k)) - \beta_k\left\langle\nabla\bar{\mathcal{L}}(\theta_k,\eta(k)),\hat{\nabla}\bar{\mathcal{L}}(\theta_k,\eta(k))+\mathcal{N}_1(\theta_k)\right\rangle + \frac{L}{2}\beta_k^2\left\|\hat{\nabla}\bar{\mathcal{L}}(\theta_k,\eta(k))+\mathcal{N}_1(\theta_k)\right\|^2$$

$$\leq \bar{\mathcal{L}}(\theta_k,\eta(k)) - \beta_k\left\langle\nabla\bar{\mathcal{L}}(\theta_k,\eta(k)),\hat{\nabla}\bar{\mathcal{L}}(\theta_k,\eta(k))\right\rangle - \beta_k\left\langle\nabla\bar{\mathcal{L}}(\theta_k,\eta(k)),\mathcal{N}_1(\theta_k)\right\rangle$$

$$+ \frac{L}{2}\beta_k^2\left[\left\|\hat{\nabla}\bar{\mathcal{L}}(\theta_k,\eta(k))\right\|^2 + \|\mathcal{N}_1(\theta_k)\|^2\right] \tag{58}$$

Taking expectations with respect to the sigma field $\mathcal{F}_k$ on both sides of (58), we obtain

$$\mathbb{E}_k\left[\bar{\mathcal{L}}\left(\theta_{k+1}\right)\right] \leq \mathbb{E}_k\left[\bar{\mathcal{L}}\left(\theta_k, \eta(k)\right)\right] - \beta_k\left\langle\nabla\bar{\mathcal{L}}\left(\theta_k, \eta(k)\right), \nabla\bar{\mathcal{L}}\left(\theta_k, \eta(k)\right) + c_3\delta\mathbf{1}_{d\times 1}\right\rangle$$

$$- \beta_k B'\mathbb{E}_k\left\|\mathcal{N}_1(\theta_k)\right\| + \frac{L}{2}\beta_k^2\left[\left\|\mathbb{E}_k\left[\widehat{\nabla}\bar{\mathcal{L}}\left(\theta_k, \eta(k)\right)\right]\right\|^2 + \frac{c_4}{\delta^2}\right] + \frac{L}{2}\beta_k^2\left\|\mathcal{N}_1(\theta_k)\right\|^2$$

$$\leq \bar{\mathcal{L}}\left(\theta_k, \eta(k)\right) - \beta_k\left\|\nabla\bar{\mathcal{L}}\left(\theta_k, \eta(k)\right)\right\|^2 + c_3\delta\beta_k\mathbb{E}_k\left\|\nabla\bar{\mathcal{L}}\left(\theta_k, \eta(k)\right)\right\|_1 - B'\beta_k\frac{B_{10}\beta_k}{\delta}$$

$$+ \frac{L}{2}\beta_k^2\left[\left\|\nabla\bar{\mathcal{L}}\left(\theta_k, \eta(k)\right) + c_3\delta\mathbf{1}_{d\times 1}\right\|^2 + \frac{c_4}{\delta^2}\right] + \frac{L}{2}\beta_k^2\frac{B_{14}\beta_k^2}{\delta^2} \tag{59}$$

$$\leq \bar{\mathcal{L}}\left(\theta_k, \eta(k)\right) - \beta_k\left\|\nabla\bar{\mathcal{L}}\left(\theta_k, \eta(k)\right)\right\|^2 + c_3\delta\beta_k\mathbb{E}_k\left\|\nabla\bar{\mathcal{L}}\left(\theta_k, \eta(k)\right)\right\|_1 - \frac{B'B_{10}\beta_k^2}{\delta}$$

$$+ \frac{L}{2}\beta_k^2\left[\left\|\nabla\bar{\mathcal{L}}\left(\theta_k, \eta(k)\right)\right\|^2 + 2c_3\delta\mathbb{E}_k\left\|\nabla\bar{\mathcal{L}}\left(\theta_k, \eta(k)\right)\right\|_1\right] + \frac{L}{2}\beta_k^2\left[dc_3^2\delta^2 + \frac{c_4}{\delta^2}\right] + \frac{L}{2\delta^2}B_{14}\beta_k^4$$

$$\leq \bar{\mathcal{L}}\left(\theta_k, \eta(k)\right) - \left(\beta_k - \frac{L}{2}\beta_k^2\right)\left\|\nabla\bar{\mathcal{L}}\left(\theta_k, \eta(k)\right)\right\|^2 + c_3\delta B'\left(\beta_k + L\beta_k^2\right) - \frac{B'B_{10}\beta_k^2}{\delta} +$$

$$\frac{L}{2}\beta_k^2\left[dc_3^2\delta^2 + \frac{c_4}{\delta^2}\right] + \frac{L}{2\delta^2}B_{14}\beta_k^4, \tag{60}$$

The 1st inequality follows from (56), (57) in Lemma 10, and from Lemma 8 . In the above, $-\|y\|_1 \leq \sum_{i=1}^d y_i$ for any $d$-vector $y$, is used to get the inequality in (59). The last inequality follows from the fact that $\left\|\nabla\bar{\mathcal{L}}\left(\theta_k, \eta(k)\right)\right\|_1 \leq B'$ by Lemma 9. Now, re-arranging the terms,

$$\left\|\nabla\bar{\mathcal{L}}\left(\theta_k, \eta(k)\right)\right\|^2 \leq \frac{2}{\beta_k(2 - L\beta_k)}\left[\bar{\mathcal{L}}\left(\theta_k, \eta(k)\right) - \mathbb{E}_k\bar{\mathcal{L}}\left(\theta_{k+1}\right) + c_3\delta\left(\beta_k + L\beta_k^2\right)B'\right]$$

$$+ \frac{L\beta_k^2}{\beta_k(2 - L\beta_k)}\left[dc_3^2\delta^2 + \frac{c_4}{\delta^2}\right] + \frac{2\beta_k^2}{\beta_k(2 - L\beta_k)}\left[\frac{LB_{14}}{2\delta^2}\beta_k^2 - \frac{B'B_{10}}{\delta}\right]$$

Now, as in (Wu et al., 2020), we sum up the inequality above for $k = \iota_m$ to $m$, take expectations, divide by $(1 + m - \iota_m)$ both sides and assume $m > 2\iota_m - 1$. We now obtain

$$\frac{1}{1 + m - \iota_m}\sum_{k=\iota_m}^m\mathbb{E}_k\left\|\nabla\bar{\mathcal{L}}\left(\theta_k, \eta(k)\right)\right\|^2$$

$$\leq \frac{2}{1 + m - \iota_m}\sum_{k=\iota_m}^m\frac{\left(\mathbb{E}_k\bar{\mathcal{L}}\left(\theta_k, \eta(k)\right) - \mathbb{E}_k\bar{\mathcal{L}}\left(\theta_{k+1}, \eta(k)\right)\right)}{\beta_k(2 - L\beta_k)} + \frac{2}{1 + m - \iota_m}\sum_{k=\iota_m}^m c_3\delta B'\left(\frac{1 + L\beta_k}{2 - L\beta_k}\right)$$

$$+ \frac{L}{1 + m - \iota_m}\sum_{k=\iota_m}^m\frac{\beta_k}{(2 - L\beta_k)}\left[dc_3^2\delta^2 + \frac{c_4}{\delta^2}\right] + \frac{2}{1 + m - \iota_m}\sum_{k=\iota_m}^m\frac{\beta_k}{(2 - L\beta_k)}\left[\frac{LB_{14}}{2\delta^2}\beta_k^2 - \frac{B'B_{10}}{\delta}\right] \tag{61}$$

Now, we denote 1st, 2nd, 3rd and 4th terms of right-hand-side of (61) as $I_1$, $I_2$, $I_3$, and $I_4$ respectively.

In $I_1$,

$$\sum_{k=\iota_m}^{m} \frac{1}{\beta_k} * \frac{\left(\mathbb{E}_k \bar{\mathcal{L}}\left(\theta_k, \eta(k)\right) - \mathbb{E}_k \bar{\mathcal{L}}\left(\theta_{k+1}, \eta(k)\right)\right)}{(2 - L\beta_k)} \leq \sum_{k=\iota_m}^{m} \frac{1}{\beta_k} * \left(\mathbb{E}_k \bar{\mathcal{L}}\left(\theta_k, \eta(k)\right) - \mathbb{E}_k \bar{\mathcal{L}}\left(\theta_{k+1}, \eta(k)\right)\right)$$

$$= \sum_{k=\iota_m}^{m} \frac{1}{\beta_k} * \left(\mathbb{E}_k \bar{\mathcal{L}}\left(\theta_k, \eta(k)\right) - \mathbb{E}_k \bar{\mathcal{L}}\left(\theta_{k+1}, \eta(k+1)\right)\right)$$

$$+ \sum_{k=\iota_m}^{m} \frac{1}{\beta_k} * \left(\mathbb{E}_k \bar{\mathcal{L}}\left(\theta_{k+1}, \eta(k+1)\right) - \mathbb{E}_k \bar{\mathcal{L}}\left(\theta_{k+1}, \eta(k)\right)\right)$$

$$= \sum_{k=\iota_m}^{m} \left(\frac{1}{\beta_k} - \frac{1}{\beta_{k-1}}\right) \mathbb{E}_k \left[\bar{\mathcal{L}}\left(\theta_k, \eta(k)\right)\right] \frac{1}{\beta_{\iota_m - 1}} \mathbb{E}_k \left[\bar{\mathcal{L}}\left(\theta_{\iota_m}, \eta(\iota_m)\right)\right] - \frac{1}{\beta_m} \mathbb{E}_k \left[\bar{\mathcal{L}}\left(\theta_{m+1}, \eta(m+1)\right)\right]$$

$$+ \sum_{k=\iota_m}^{m} \frac{1}{\beta_k} * \left[(B_g + B_\nu) \sum_{q=1}^{N} |\eta_q(k+1) - \eta_q(k)|\right]$$

$$\leq \sum_{k=\iota_m}^{m} \left(\frac{1}{\beta_k} - \frac{1}{\beta_{k-1}}\right) B_h + \frac{1}{\beta_{\iota_m - 1}} B_h - \frac{1}{\beta_m} B_h + N(B_g + B_\nu)^2 \sum_{k=\iota_m}^{m} \frac{\zeta_k}{\beta_k}$$

$$\leq B_h \left[\sum_{k=\iota_m}^{m} \left(\frac{1}{\beta_k} - \frac{1}{\beta_{k-1}}\right) + \frac{1}{\beta_{\iota_m - 1}}\right] + B_{12} \sum_{k=\iota_m}^{m} (1+k)^{\sigma_6 - \sigma_4}$$

$$\leq 2B_h \beta_m^{-1} + B_{12} \frac{\{m - \iota_m + 1\}^{1 - \sigma_6 + \sigma_4}}{1 - \sigma_6 + \sigma_4}$$

$$= 2B_h \beta_m^{-1} + B_{13}\{m - \iota_m + 1\}^{1 - \sigma_6 + \sigma_4},$$

In the above, the 1st inequality is due to $\beta_k \leq \frac{1}{L}$. The 2nd inequality holds due to $|\mathbb{E}_k[\bar{\mathcal{L}}(\theta_k), \eta(k)]| \leq B_h$ as mentioned above. The second last inequality is from

$$\sum_{k=0}^{m-\iota_m} (1+k)^{-(\sigma_4 - \sigma_6)} \leq \int_0^{m-\iota_m+1} y^{-(\sigma_4 - \sigma_6)} \, dy = \frac{(m - \iota_m + 1)^{1 - (\sigma_4 - \sigma_6)}}{1 - (\sigma_4 - \sigma_6)}$$

and in the equality $B_{13} = B_{12}/\sigma_4 > 0$.

From $I_2$,

$$\frac{2}{1 + m - \iota_m} \sum_{k=\iota_m}^{m} c_3 \delta B' \left(\frac{1 + L\beta_k}{2 - L\beta_k}\right) \leq \frac{2}{1 + m - \iota_m} \sum_{k=\iota_m}^{m} c_3 \delta B'(1 + L\beta_k)$$

$$\leq \frac{2}{1 + m - \iota_m} \sum_{k=\iota_m}^{m} 2c_3 \delta B' \beta_k \leq B_{15} \beta_m = \mathcal{O}\left(\frac{1}{m^{1/2}}\right)$$

In the above the 1st and 2nd inequality is due to $\beta_k \leq \frac{1}{L}$ and $B_{15} > 0$ some constant term.

From $I_3$ we get,

$$\frac{L}{1 + m - \iota_m} \sum_{k=\iota_m}^{m} \frac{\beta_k}{(2 - L\beta_k)} \left[dc_3^2 \delta^2 + \frac{c_4}{\delta^2}\right] \leq \frac{L}{1 + m - \iota_m} \sum_{k=\iota_m}^{m} \beta_k \left[dc_3^2 \delta^2 + \frac{c_4}{\delta^2}\right] \leq B_{16} \beta_m = \mathcal{O}\left(\frac{1}{m^{1/2}}\right)$$

In the above, first inequality is due to $\beta_k \leq \frac{1}{L}$, and $B_{16} > 0$.

Further, from $I_4$

$$\frac{2}{1+m-\iota_m} \sum_{k=\iota_m}^{m} \frac{\beta_k}{(2-L\beta_k)} \left[ \frac{LB_{14}}{2\delta^2} \beta_k^2 - \frac{B'B_{10}}{\delta} \right]$$

$$\leq \frac{2}{1+m-\iota_m} \sum_{k=\iota_m}^{m} \beta_k \left[ \frac{LB_{14}}{2\delta^2} \beta_k^2 - \frac{B'B_{10}}{\delta} \right] \leq B_{17}\beta_m^3 = \mathcal{O}(\frac{1}{m^{3/2}})$$

In the above, first inequality is due to $\beta_k \leq \frac{1}{L}$, and $B_{17} > 0$.

Now from (61)

$$\min_{0 \leq k \leq m} \mathbb{E} \left\| \nabla \bar{\mathcal{L}}(\theta_k, \eta(k)) \right\|^2 = \frac{1}{1+m-\iota_m} \sum_{k=\iota_m}^{m} \mathbb{E}_k \left\| \nabla \bar{\mathcal{L}}(\theta_k, \eta(k)) \right\|^2$$

$$\leq \frac{4B_h \beta_m^{-1}}{1+m-\iota_m} + B_{13} \frac{\{m-\iota_m+1\}^{1-\sigma_6+\sigma_4}}{1+m-\iota_m} + \mathcal{O}(\frac{1}{m^{1/2}}) + \mathcal{O}(\frac{1}{m^{3/2}})$$

$$\leq \frac{4B_h \max\{L, \{1+m\}^{1/2}\}}{1+m-\iota_m} + \frac{B_{13}}{\{m-\iota_m+1\}^{\sigma_6-\sigma_4}} + \mathcal{O}(\frac{1}{m^{1/2}})$$

$$= \mathcal{O}(\frac{1}{m^{1/2}}) + \mathcal{O}(\frac{1}{m^{1/2}}) + \mathcal{O}(\frac{1}{m^{1/2}}) = \mathcal{O}(\epsilon^{-2})$$

$\square$

