# OpenReview forum: "Actor-only and Safe-Actor-only REINFORCE Algorithms with Deterministic Update Times"
_TMLR — Rejected by TMLR_

### Review · Reviewer_geDY · 2025-04-27

**Summary Of Contributions:**

The paper derives a on-policy algorithm that achieves better finite-time complexity bound than existing methods---this is done by assuming simulation access to obtain two samples from the current policy and a perturbed policy, and taking their finite differences in the rewards. The paper provides a deterministic update schedule and the corresponding step sizes to obtain this improved rate. The paper further extends this algorithm to the constrained MDP setting where the algorithm also enjoys the improved rate under this setting. Finally, the paper conducts numerical analyses on grid-world environments and claim to outperform SOTA results.

**Audience:**

Yes

**Broader Impact Concerns:**

As far as I know there isn't any broader impact concerns.

**Claims And Evidence:**

No

**Requested Changes:**

- Given that I have a lot of confusion about the writing, I believe the paper should clearly indicate the definitions and assumptions used for each lemma/proposition/theorem/algorithm. Since this is mostly a theoretical paper these are very important for verifying correctness.
- The paper should include the implementation details for reproducible results, and also change the claim that it achieves SOTA performance.

**Strengths And Weaknesses:**

**Strengths**
- The proposed algorithms have a deterministic update schedule which does not require termination of the trajectory like REiNFORCE.
- The paper provides detailed asymptotic and finite-time convergence bounds for derived algorithms under common assumptions such as linear policies, uniform ergodicity, etc.

**Major Comments**
I find the paper to be fairly difficult to read, perhaps it is missing specific notations/descriptions/definitions of variables. I also believe not all assumptions/limitations are explicitly indicated. For example:
- In Algorithm 1 and 2:
	- What are $\sigma''$ and $\sigma'$? Are they manually defined? Likewise for $\sigma_4, \sigma_5$, and $\sigma_6$. Also this seems to be inconsistent and adds confusion
	- $m$ is never updated in the algorithm, should that happen after line 10?
	- Do we sample $\Delta$ at every timestep? What's $\Delta_{m, i}$?
	- Which current state $s$ do we continue with?
- Section 4:
	- Generally the description of Algorithm 1 is difficult to understand, and it barely provides any high-level intuition.
		- E.g., $m$ seems to mean the number of total updates, where samples from $n_{m}$ to $n_{m+1}$ are used for the $m + 1$'th update.
		- Page 5, the filtration $\sigma$ is mentioned but never again and can be confusing along with other variables such as $\sigma'$ and $\sigma_4$.
- Section 5:
	- For theorem 2, is $\eta$ fixed in the analysis? How does this apply to changing $\eta$ as the algorithm changes it? Although proposition 1 indicates that $\eta$ converges asymptotically as well, this result seems to be independent of theorem 2.
- Section 6:
	- Theorem 3 is not a last-iterate convergence, is this easily extendable?
	- I think there should be some remarks about what lemmas 3-6 are indicating, for example:
		- Lemma 4 should mean that with large $m$ we can expect $\mathcal{N}$ to shrink.
		- Under these lemmas which specific assumptions do we need?
- Remark 1
	- This statement is very confusing, in Algorithm 1 we have $n$ that is the number of transitions while $m$ is, I suppose, the number of updates. Does $T$ correspond to the former or the latter?
	- Also, does this algorithm do better under exact same assumptions? It seems like the simulation access might play a role here, can the paper elaborate how this helps with improved rates over methods like actor critic?

There are few experimental concerns:
- Hyperparameters and implementation details are not provided, e.g., tabular vs linear softmax policies, step sizes for different algorithms, etc.
- The conclusion indicates that the proposed algorithm is superior, but non of the results are statistically significant.
- The analysis on computation time is unclear. Is this evaluated based on the update time, or including the rollout? If so, the double-simulation should be a factor to consider. In some applications transitioning takes more time than expected.
- Given that the algorithm assumes simulation access and samples two transitions per state, is it really fair to compare the algorithms with same number of timesteps?

**Typos/Writing**
- Page 1, introduction, MDP is used before it is defined.
- Page 1, introduction, after the word "robotics" there should be a space before reference.
- Page 2, related work, TD is used before it is defined.

**Questions**
- Regarding remark 1, would having more samples for the current state speed up convergence, perhaps in the constant term?
- How do these results compare against model-based methods? Are the proposed algorithms minimax optimal?

---

> ### Author Response · Authors · 2025-05-14
> **Response to the Reviewer geDY**
>
> Response to the Reviewer geDY:
>
> In Algorithms 1 and 2:
> 1. For better clarity and consistency, updated and clearly defined the notations in the revised paper. Further, added a clear description and usage in Appendix A.
>
> $\sigma'$, $\sigma''$ in the earlier version (i.e., $\sigma'''$, $\sigma''$ in the revised version) are positive constants that can take values such that $0<\sigma''<\sigma'''\leq 1$ (see line 1, Algorithm 1). Similarly, $\sigma_4$, $\sigma_5$, $\sigma_6$ can take values such that $0<\sigma_4<\sigma_5<\sigma_6\leq1$ (see line 1, Algorithm 2).
>
> Yes, we need to manually define values of $\sigma'''$, $\sigma''$, $\sigma_4$, $\sigma_5$, and $\sigma_6$.
>
> For example, we choose $\sigma'''=1$, $\sigma''=0.5$ (see Theorem 3), $\sigma_4 = 0.49$, $\sigma_5=0.99$ and $\sigma_6=1$ (see Theorem 4).
>
> 2. Yes, $m$ is updated after line 10, see line 11, Algorithm 1.
>
> 3. We now revise as we sample perturbation vector $\Delta$ at every $m$th instant, see 3rd para, page 5, section 4.1.
>
> Now, updated notation $\Delta_{m,i}$ as $\Delta_i(m)$ (defined on page 5).
>
> 4. we update current state $s$ as next state $s'$, received using $(s,a \sim \pi^{\theta})$ (see line 14, Algorithm 1).
>
> Section 4:
>
> 1. Yes, while $m$ keeps track of the number of times the parameter $\theta$ is updated, it also denotes that the actual time instant of the $m$th update is $n_m$ (see 4th para, page 5).
>
> 2. For better clarity, removed $\sigma$ symbol, and revised on page 5 as 'sigma-field generated by ...'. Note that this sigma-field is used in the finite-time analysis (pages 9, 11).
>
> Section 5:
>
> Yes, $\eta$ is fixed in the analysis of Theorem 2 as $\eta$ is updated in the slower timescale ($\zeta_n < \alpha_n, \forall n \geq 0$) than $\theta$-update (see page 8 for discussions).
>
> Section 6:
>
> 1. In Theorem 3, we provide the finite-time analysis of our Algorithm 1, whereas in Theorem 1, we provide the asymptotic convergence of Algorithm 1. The detailed proof of Theorem 3 is provided in Appendix A.2.1.
>
> 2. We have introduced Remark 1 in Section 6.1 to address the significance and required assumptions for Lemmas 3-6.
>
> Remark 1: (in the earlier version, that is remark 3 in the revised version)
>
> 1. For better clarity, we added more technical details on page 5. Even if in the derivation of finite-time analysis of Algorithm 1, we use $m$, $T$ corresponds to both $n$ and $m$.
>
> 2. Yes, we followed standard assumption Assumption 3, as that is considered in the literature.
>
> Our Algorithm 1 achieves $ \mathcal{O}(\epsilon^{-0.5})$ times better sample complexity than the AC algorithm. Let's say, $\epsilon=0.02$, then the AC algorithm requires 17677 samples, whereas our algorithm requires only 2500 samples.
>
> Further, we have compared our finite-time complexity with five more works in the literature. In Table 1, Remarks 3 and 4, we mention the novelty of our proposed algorithms in relation to other algorithms in the literature.
>
> Experimental concerns:
>
> 1. Hyperparameters and implementation details are updated on the 1st para, section 7, page 12 in the revised paper.
>
> 2. In the Conclusions, we have updated SOTA as considered. We have now shown the results of experiments on another two standard RL benchmark environments, 'CartPole' and 'Acrobot', in addition to different 'Grid World' environments (see Table 4, Figure 5). These results confirmed that the performance of our algorithms is generalizable across environments and better than popular existing algorithms.
>
> 3. In Table 3, the obtained computation time is the total time needed to execute $n=1,00,000$ iterations of our algorithms. This time is the total time, including the time to get the two simulations, in our proposed algorithms.
>
> 4. We showed computation time (see Table 3) for all considered algorithms, and our algorithm showed that it takes less time than other algorithms. So, this is fair to compare with the considered algorithms.
>
> Typos/Writing:
>
> All are corrected.
>
> Questions:
>
> 1. More samples at the current state can help to improve the constant term, but not the convergence rate [3].
>
> $[3]$ ``Finite-Time Convergence and Sample Complexity of Actor-Critic Multi-Objective Reinforcement Learning'' by Zhou et al., ICML 2024, article 2562, pages 61913 - 61933.
>
> 2. In the model-based setting, one models/learns transition probabilities that in themselves are expensive. Our algorithms directly work with data and will be easier to deploy in real-world applications.
> Single-agent setting is considered in our work, so minimax optimality is not applicable. In the future, one can possibly extend our work to multi-agent settings where proving minimax optimality will be useful.
>
> Requested Changes:
> 1. The revised paper now clearly indicates the definitions and assumptions used for each lemma/proposition/theorem/algorithm.
> 2. See point no. 1 of Experimental concerns.
>
> In the revised paper, we updated 'state-of-the-art (SOTA) algorithms' as 'popular algorithms'. or 'considered algorithms'

---

### Review · Reviewer_6fJC · 2025-04-30

**Summary Of Contributions:**

This work proposes actor only algorithms for standard MDP and constrained MDP, respectively. Their algorithms update actor parameter after increasing deterministic instants using a Simultaneous Perturbation Stochastic Approximation (SPSA) based approach. Asymptotic convergence and finite-time convergence analysis are provided to show the proposed algorithms improve upon existing methods in terms of convergence rate.

**Audience:**

Yes

**Broader Impact Concerns:**

N.A.

**Claims And Evidence:**

Yes

**Requested Changes:**

1. The citations look weird, they are mixed into sentences. I am not sure if this is related to the choice of \citet or \citep.

2. How strong is assumption 1? Does it mean with this assumption, exploration is no longer necessary (or guaranteed)?

3. In algorithm 1, line 5 and 6 use simulator to generate next state and reward in parallel. But what is the current state $s$ update? Is it $s'$?

4. What is the $\hat{\Lambda}$ in (11)?

5. On page 7, (34) should be (3)?

6. The procedure of Algorithm 1 and 2 is clear, but can you explain the design and intuition of why it works? For example, how the deterministic update times and two/three timescale help to find the optimal parameter?

7.  In the introduction, you mention "In this paper, we first present an Actor-only algorithm (AOA) that works with a single update recursion (instead of two recursions commonly used in the AC method) and works with linear function approximation." Can you explain where is the linear function approximation in your algorithm and analysis? The only function approximation I can tell is the policy class parameterized by $\theta$.

8. The authors assume finite state and action spaces in this work, do you have any idea about how does your method extend to infinite state and action spaces?

9. For the finite time analysis results, the convergence is in the order of $O(\epsilon^{-2})$. How do the number of states and actions affect the convergence rate? Is there any other key factor that is hided in this bound?

10. It would be great if the authors can explicitly discussion the novelty in their algorithm design and theoretical analysis.

**Strengths And Weaknesses:**

This work provides new algorithms and improved convergence results. The delivery of the algorithm is pretty clear, but the discussion on the intuition and novelty have space to improve. Please see requested changes for more details.

---

> ### Author Response · Authors · 2025-05-14
> **Response to the Reviewer 6fJC**
>
> Response to the Reviewer 6fJC:
>
> Requested Changes:
>
> 1. Updated citations appropriately by using $\backslash$citep.
>
> 2. Assumption 1 is a standard assumption considered in the reinforcement learning literature.
>
> By Assumption 1, the Markov chain is irreducible, aperiodic, and positive recurrent. Under this assumption, the Markov chain has a unique stationary distribution $d_{\pi}$ (see 1st para, page 5 of revised paper).
>
> Assumption 1 does not say anything about the exploration of the RL agent.
>
> 3. Yes, now we mention ``Update the current state $s$ as the next state $s'$, i.e., $s \leftarrow s'$'' in Line 14, Algorithm 1, i.e., we update current state as next state $s'$, received using $(s,a \sim \pi^{\theta})$.
>
> 4. In (11), $\hat \Lambda: \mathbb{R} \rightarrow [0,B_{\eta}]$ denotes the projection of $z$, $\hat \Lambda(z)=\max(0,\min(z,B_{\eta})$, for any $z \in \mathbb{R}$, where $B_{\eta} < \infty$ is a large positive constant. $\hat \Lambda$ ensures boundedness of Lagrange parameters $\eta_q$.
>
> 5. (34), page 7 of the earlier version is correct. However, for more clarity, at 5th para, page 5, revised version, we mentioned that ``See (3) for $J_{\pi^{\theta}}$ but for the parameterized policy $\pi^{\theta}$, i.e., see (34) in the Appendix.''.
>
> 6. The design and intuition of algorithms 1 and 2 are revised in section 4.
>
> In Monte-Carlo PG methods, the update happens after the end of each episode, and episode lengths are random, not deterministic, which can lead to unpredictable or sparse parameter updates. Our algorithms resolve the unpredictability of parameter update that lies in the Monte-Carlo PG methods. Our proposed algorithms perform parameter updates after increasing deterministic instants, unlike the regular Monte-Carlo PG methods.
> In section 4.1 of the revised paper, we mentioned that the proposed algorithm has a single update recursion, update of parameter $\theta$, that involves step-size ${\alpha_n, n\geq 0}$. The, update epochs, $n_m \geq 0$, or alternatively $m \geq 0$ are obtained from two sets of step-sizes ${\alpha_n, n\geq 0}$, and ${\beta_n, n\geq 0}$, respectively. See line no. 4 of Algorithm 1 to get the update epochs or update instants.
>
> Further, regarding obtaining the convergence and optimal parameter in algorithm 2, we added intuitions and explanations after eq. (17), page 8, section 5.
>
> 7. In this paper, the goal is to receive the optimal policy parameter, and thus we have used linear softmax policy parameterization (see eq. (6)), alternatively, linear function approximation to the parameterized policies. Function approximation is not required in any other place in this work.
> For better clarity, we mentioned in the revised paper that ``We consider linear soft-max policy parameterizations (see eq. (6) as we consider linear function approximation in this work.'' in section 4.1.
>
> 8. Our methods can be extended to infinite state and action spaces. However, to prove the stability and convergence for the case of infinite state and action spaces, extra assumptions will be needed. In future, one can extend our work to the infinite state and action spaces.
>
> 9. The number of states and actions do not affect the convergence rate, as model-free data driven approach with function approximation is used in this work.
>
> No hidden key factor is there in our convergence rate $\mathcal{O}(\epsilon^-{2})$. Further, we have shown the detailed analysis of our finite-time convergence rates, see pages 24-25, Appendix A.2.1.
>
> 10. We have improved the exposition of the contributions of our work by more clearly describing the novelty.
> At the end of the introduction section in the revised paper, we mentioned our specific contributions, such as
>
> Our algorithms resolve the unpredictability of parameter update that lies in the Monte-Carlo PG methods. Our proposed algorithms perform parameter updates after increasing deterministic instants, unlike the regular Monte-Carlo PG methods, where the update happens after the end of each episode (episode lengths are random, not deterministic).
>
> We consider two different MDP settings - with and without constraints, and propose RL algorithms for both settings.
>
>  Both of our algorithms, AOA (involves two timescales) and SAOA (involves three timescales), are model-free RL algorithms and, in fact, versions of PG and constrained PG algorithms, respectively.
>
>  We provide the convergence guarantees of our proposed algorithms by performing both asymptotic and non-asymptotic or finite-time analysis. To the best of our knowledge, there are no other two-timescale and three-timescale algorithms that provide this sample complexity while remaining almost surely stable and convergent.
>
>  We also provide empirical results, including regular and safe navigation of the RL agent in different standard environments, that demonstrate the effectiveness of the theoretical results.

---

### Review · Reviewer_3rZd · 2025-05-02

**Summary Of Contributions:**

This work proposed a policy optimization method for solving average-reward MDP and constrained MDP problems without directly using value functions. The authors cast the problem into a standard first-order stochastic analysis framework where the bias and total expected error terms are bounded by using techniques involving two simulations during the data collection time.

**Audience:**

Yes

**Broader Impact Concerns:**

Not applicable.

**Claims And Evidence:**

No

**Requested Changes:**

1. More related work discussions. I find the scope and range of literature citated are severely lacking.
2. Better presentation of the results. I find the notation usage are hard to read and follow.
3. Simple gridworld experiments can only be used as toy examples, regardless of scale.

**Strengths And Weaknesses:**

1. The writing of this paper needs significant improvement. Besides numerous grammatical error, I find the inappropriate usage of citation format, presentation of assumptions, and the casual use of undefined notations make this paper very difficult to read, even though the main idea of this paper-using stochastic gradient descent analysis framework-are not novel or interesting.
2. Moreover, I found this work lack serious treatment of the theoretical RL community, where a plethora of average-reward MDP and/or constrained MDP papers have been proposed such as, to name a few:

[1] Jin, Yujia, and Aaron Sidford. "Towards tight bounds on the sample complexity of average-reward MDPs." International Conference on Machine Learning. PMLR, 2021.

[2] Zhang, Zihan, and Qiaomin Xie. "Sharper model-free reinforcement learning for average-reward markov decision processes." The Thirty Sixth Annual Conference on Learning Theory. PMLR, 2023.

[3] Grand-Clément, Julien, and Marek Petrik. "Reducing blackwell and average optimality to discounted mdps via the blackwell discount factor." Advances in Neural Information Processing Systems 36 (2023): 52628-52647.

[4] Li, Tianjiao, Feiyang Wu, and Guanghui Lan. "Stochastic first-order methods for average-reward markov decision processes." Mathematics of Operations Research (2024).

[5] Tiapkin, Daniil, and Alexander Gasnikov. "Primal-dual stochastic mirror descent for MDPs." International Conference on Artificial Intelligence and Statistics. PMLR, 2022.

Notbly, the claimed convergence rate has been well established for both generative model setting and the markovian model setting (see [4]). In terms of the iteration complexity, I don't see how this work can improve from previous works.
3. Thirdly, theorem 3, the main theorem of this paper, assumes that the objective function, the average-reward, is $L$-smooth w.r.t to policy parameters $ \theta $. This is an assumption I found extremely unusual and quite concerning. Can the author elaborate how this condition can be achieved through rigorous arguments?
4. The algorithm framework clearly requires a generative model, in the sense that the algorithm needs to reset one of the simulators into a arbitrary state to make sure the state are synced between the two simulators. This contradicts with the Markovian model claim. Additionally, the policy also needs a copy to obtain two actions at the same time.
5. Can the authors either use $r$ or $c$ and make it consistant?
6. What is $J(n)$?
7. I find the comparison against SAC, a discounted method, is unfitting in the experiment section. More of the fact that SAC is not a theoretical work. The experiments of using gridworld are not convincing. Neither the standard actor-critic method or SAC is the state of the art method. The author seems to compare the standard RL method with the entropy regularized RL methods.

---

> ### Author Response · Authors · 2025-05-14
> **Response to the Reviewer 3rZd**
>
> Response to the Reviewer 3rZd:
>
> Strengths And Weaknesses:
>
> 1. (a) The writing of the paper improved significantly in the revised version. We have corrected grammatical errors and used the appropriate citation format.
>
> (b) We have considered the standard assumptions as in the literature and explicitly mentioned assumptions used for each lemma, proposition, and theorem (see sections 5 and 6).
>
> (c) To improve the readability of the paper, we have revised notations, added descriptions, and made them consistent throughout the revised paper.
>
> (d) We have improved the exposition of the contributions of our work by more clearly describing the novelty.
> At the end of the introduction section in the revised paper, we mentioned our specific contributions (see section 1, page 2).
>
> 2. We have enriched the literature survey by adding five more references (see sections 2 and 6).
> We clearly mentioned the contributions and novelty of our work compared to theoretical RL works such as [4] in section 2 and Remark 3 and 4, in section 6.
>
> In the 2nd para of section 2, we discussed the work done in [4], and pointed out the limitations of that work. Note that our proposed work overcomes all those limitations. In Remark 3, section 6, we mention that our AOA algorithm achieves the sample complexity as good as in [4] while having deterministic update instants, unlike [4]. Furthermore, our AOA not only provides better sample complexity than majority of the considered existing algorithms but also remains almost surely stable and convergent (see Theorem 1), whereas almost sure convergence is not guaranteed in [4]. Further, our SAOA is for the constrained RL unlike [4].
>
> [4] as in the Reviewer 3rZd 's comment.
>
> 3. In the revised paper, we introduced Remark 2 to show that $\bar J$ is $L$-smooth (see page 10, section 6.1).
>
> 4. (a) The algorithm framework follows the Markovian model. In the revised paper, in the Algorithm 1 (see line no. 14)  and Algorithm 2 (see line no. 20), we mentioned that ``Update the current state $s$ as the next state $s'$, i.e., $s \leftarrow s'$'', that is we set next state $s'$, received using $(s, a \sim \pi^{\theta})$ as current state $s$. The algorithm does not require a generative model and does not need to reset one of the simulators into an arbitrary state.
>
> (b) As we mentioned on line no. 6, Algorithm 1, we need a copy of the parameter $\theta$ to get the perturbed value $\theta +\delta\Delta$ and sample action, but that does not affect the Markov property.
>
> 5. Throughout the analysis, we consistently use $c$ and for the better clarity explicitly mention in section 5.1 that, ``For purposes of the remaining analysis, we alternatively consider a cost minimization problem (instead of a reward maximization) wherein we set $c(j)=-r(j)$, $\forall j$.''
>
> 6. In this algorithm, the long-term average reward $J(n)$ at time instant $n$ is calculated iteratively (see line 13, Algorithm 1). As $\beta_n > \alpha_n, \forall n \geq 0$, that is, $J(n)$ evolves in the faster timescale than $\theta$ (evolves in the slower timescale), in the long run
> $J(n)$ converges to $J_{\pi^{\theta}}$.
>
> 7. (a) To ensure the fairness of the comparison among algorithms, we calculate and demonstrate the computation time (in seconds) for all algorithms (see Table 3). This table shows that our proposed AOA consumes less computation time than SAC. As SAC is a popular RL algorithm, our experimental results show that our algorithm obtains better average reward and better computational time complexity, which ensures the effectiveness of our algorithm.
>
> (b) We have now shown the results of experiments on another two standard RL benchmark environments, 'CartPole' and 'Acrobot' (used in the recent RL literature, available in the OpenAI gym), in addition to different 'Grid World' environments that were shown in the earlier version. We have introduced a new figure, Figure 5, and a new table, Table 4, to demonstrate the performance of our approach in the CartPole and Acrobot environments in the revised version (see pages 12,14,15). These results confirmed that the performance of our algorithms is generalizable across environments.
>
> In the revised paper, we updated 'state-of-the-art (SOTA) algorithms' as 'popular algorithms'.
>
> Requested Changes:
>
> 1. We have enriched the literature survey by adding all references [1]-[5] suggested by the reviewer in sections 2 and 6.
>
> 2. We have enhanced the presentation of the result by revising Table 1, Remarks 3 and 4. Further, we introduced Remarks 1 and 2 to enhance the clarity of the results.
>
> To improve the readability and understanding of the paper, we have revised notations, added descriptions, and made them consistent throughout the revised paper.
>
> [1]-[5] in the Reviewer 3rZd 's comment.
>
> 3. We now perform experiments on two more environments, ‘CartPole’ and ‘Acrobot’ (see 7.(b) in the above).

---

### Decision · Action_Editor_ndes · 2025-06-20

**Recommendation:** Reject

**Additional Comments:**

I provided the requested revisions in the explanation for the "Evidence" criterion, above.

**Audience:**

Yes

**Audience Explanation:**

The reviewing team agrees that the audience of TMLR would be interested in the findings of the paper.

**Claims And Evidence:**

No

**Claims Explanation:**

This paper proposes a model-free RL algorithm to find approximately stationary policies where the problem is finding a policy that maximizes the long-term average reward. In particular, the authors show that to get the policy $\pi_\theta$, parameterized by $\theta$, with $\|\nabla \bar{J}(\theta)\|^2 \leq \epsilon$, one needs the sample complexity $O(\epsilon^{-2})$ where $\bar{J}$ is the average reward in Eq. (12). The authors then also show the extension of these ideas to a constrained problem.

Reviewers geDY and 3rZd provided in-depth reviews and they still have many remaining concerns that stop me from recommending acceptance. I also agree with these concerns and I have some more of my own.

Some of these concerns include:

- Both reviewers geDY and 3rZd voiced concerns about the applicability of the new algorithm in the Markovian sampling regime, which was not addressed clearly by the authors. In particular, the reviewers do not agree that one gets two next states $s'$, $s^+$ from a state $s$ with the regular Markovian setting.

- Reviewer geDY voiced concern about the analysis of Theorem 2, which does not include the variation of $\eta$. This is only addressed in a high level comment by the authors in page 8. The reviewer is not convinced by this high level explanation. The authors are recommended to include rigorous proofs incorporating the dynamics of this parameter.

- The comparisons in the Table 1 are quite unclear. In particular, the authors are comparing their complexity results which is for getting a stationary policy with other results that get approximately optimal policies (for example, the work Li et al., 2024 finds approximately optimal policies). Moreover, the result of [Agarwal et al., 2019] should also be included and a clear comparison should be made.

[Agarwal et al., 2019] Alekh Agarwal, Sham M. Kakade, Jason D. Lee, Gaurav Mahajan, On the Theory of Policy Gradient Methods: Optimality, Approximation, and Distribution Shift, COLT 2020

- Please consider improving the presentation. For example, Eq. (20, 21, 22) looks quite difficult to parse.

Overall, the concerns above are too significant to warrant acceptance. The authors are recommended to thoroughly address the concerns of reviewers geDY and 3rZd, in addition to polishing their paper to improve presentation.

**Resubmission Of Major Revision:**

The authors may consider submitting a major revision at a later time.